# Fast and unified path gradient estimators for normalizing flows

**Lorenz Vaitl**[1*], **Ludwig Winkler**[1*], **Lorenz Richter**[2,3], **Pan Kessel**[4]

[1]Machine Learning Group, TU Berlin, [2]Zuse Institute Berlin,
[3]dida Datenschmiede GmbH, [4]Prescient Design, Genentech, Roche

## Abstract

Recent work shows that path gradient estimators for normalizing flows have lower variance compared to standard estimators for variational inference, resulting in improved training. However, they are often prohibitively more expensive from a computational point of view and cannot be applied to maximum likelihood training in a scalable manner, which severely hinders their widespread adoption. In this work, we overcome these crucial limitations. Specifically, we propose a fast path gradient estimator which improves computational efficiency significantly and works for all normalizing flow architectures of practical relevance. We then show that this estimator can also be applied to maximum likelihood training for which it has a regularizing effect as it can take the form of a given target energy function into account. We empirically establish its superior performance and reduced variance for several natural sciences applications.

## 1 Introduction

Normalizing flows (NFs) have become a crucial tool in applications of machine learning in the natural sciences. This is mainly because they can be used for variational inference, i.e., for the approximation of distributions corresponding to given physical energy functions. Furthermore, they can be synergistically combined with more classical sampling methods such as Markov chain Monte Carlo (MCMC) and Molecular Dynamics, as their density is tractable.

The paradigm of using normalizing flows as neural samplers has lately been widely adopted for example in quantum chemistry (Boltzmann generators (Noé et al., 2019)), statistical physics (generalized neural samplers (Nicoli et al., 2020)), as well as high-energy physics (neural trivializing maps (Albergo et al., 2019)). In these applications, the normalizing flow is typically trained using a combination of two training objectives: Reverse Kullback-Leibler (KL) training is used to train the model by self-sampling (see Section 2). Crucially, this training method on its own often fails in high-dimensional sampling settings because self-sampling is unlikely to probe exceedingly concentrated high probability regions of the ground-truth distribution and can potentially lead to mode collapse. As such, reverse KL training is often combined with maximum likelihood training (also known as forward KL training). For this, samples from the ground-truth distribution are obtained by standard sampling methods such as, e.g., MCMC. As these methods are typically costly, the samples are often of low number and possibly biased. The model is then trained to maximize its likelihood with respect to these samples. This step is essential for guiding the self-sampling towards high probability regions and, by extension, for successful training.

Since training normalizing flows for realistic physical examples is typically computationally challenging, methods to speed up the convergence have been a focus of recent research. To this end, *path estimators* for the gradient of the reverse KL loss have been proposed (Roeder et al., 2017; Vaitl et al., 2022a;b). These estimators focus on the parameter dependence of the flow's sampling process, also known as the sampling path, while discarding the direct parameter dependency, which vanishes in expectation. Path gradients have the appealing property that they are unbiased and tend to have lower variance compared to standard estimators, thereby promising accelerated convergence (Roeder et al., 2017; Agrawal et al., 2020; Vaitl et al., 2022a;b). At the same time, however, current path gradient estimation schemes have often a runtime that is several multiples of the standard

---

[*]Shared first authorship.

gradient estimator, thus somehow counteracting the original intention. As a remedy, recently, Vaitl et al. (2022b) proposed a faster algorithm. Unfortunately, however, this algorithm is limited to continuous normalizing flows.

Our work resolves this unsatisfying situation by proposing unified and fast path gradient estimators for all relevant normalizing flow architectures. Notably, our estimators are between 1.5 and 8 times faster than the previous state-of-the-art. Specifically, we a) derive a recursive equation to calculate the path gradient during the sampling procedure. Further, for flows that are not analytically invertible, we b) demonstrate that implicit differentiation can be used to calculate the path gradient without costly numerical inversion, resulting in significantly improved system size scaling.
Finally, we c) prove by a change of perspective (noting that the forward KL divergence in data space is a reverse KL divergence in base space) that our estimators can straightforwardly be used for maximum likelihood training. Crucially, the resulting estimators allow us to work directly on samples from the target distribution. As a result of our manuscript, path gradients can now be used for all widely used training objectives — as opposed to only objectives using self-sampling — in a unified and scalable manner.
We demonstrate the benefits of our proposed estimators for several normalizing flow architectures (RealNVP and gauge-equivariant NCP flow) and target densities with applications both in machine learning (Gaussian Mixture Model) as well as physics ($U(1)$ gauge theory, and $\phi^4$ lattice model).

## 1.1 RELATED WORKS

Pathwise gradients take the sampling path into account and are well established in doubly stochastic optimization, see e.g. L'Ecuyer (1991); Jankowiak & Obermeyer (2018); Parmas & Sugiyama (2021). The present work uses path gradient estimators, a subset of pathwise gradients, originally proposed by Roeder et al. (2017) in the context of reverse KL training of Variational Autoencoders (VAE), which is motivated by only using the sampling path for computing gradient estimators and disregarding the direct parameter dependency. These were subsequently generalized by Tucker et al. (2019); Finke & Thiery (2019); Geffner & Domke (2021a;b) to generic VAE self-sampling losses.
There has been substantial work on reducing gradient variance not by path gradients but with control variates, for example in Miller et al. (2017); Kool et al. (2019); Richter et al. (2020); Wang et al. (2023). For an extensive review on the subject, we refer to Mohamed et al. (2020).
Bauer & Mnih (2021) generalized path gradients to score functions of distributions which do not coincide with the sampling distribution in the context of hierarchical VAEs. As we will show, our fast path gradient for the forward KL training can be brought into the same form. However, only our formulation allows the application of a fast estimation scheme for NFs and establishes that forward and reverse path gradients are closely linked.
Path gradients for normalizing flows have recently been studied: Agrawal et al. (2020) were the first to apply path gradients to normalizing flows as part of a broader ablation study. However, their algorithm has double the runtime and memory constraints as it requires a full copy of the neural network. Vaitl et al. (2022a) proposed a method that allows path gradient estimation for any explicitly invertible flow at the same runtime cost as Agrawal et al. (2020) but half the memory footprint. They also proposed an estimator for forward KL training which is however based on reweighting and thus suffers from poor system size scaling, while our method works on samples from the target density. For the rather restricted case of continuous normalizing flows, Vaitl et al. (2022b) proposed a fast path gradient estimator. Our proposal unifies their method in a framework which applies across a broad range of normalizing flow types.

## 2 NORMALIZING FLOWS

A normalizing flow is a composition of diffeomorphisms

$$x = T_\theta(x_0) := T_{L,\theta_L} \circ \cdots \circ T_{1,\theta_1}(x_0), \tag{1}$$

where we have collectively denoted all parameters of the flow by $\theta := (\theta_1, \ldots, \theta_L)$. Since diffeomorphisms form a group under composition, the map $T_\theta$ is a diffeomorphism as well.
Samples from a normalizing flow can be drawn by applying $T_\theta$ to samples from a simple base density $x_0 \sim q_0$ such as $q_0 = \mathcal{N}(\mathbf{0}, \mathbb{1})$. The density of $x = T_\theta(x_0)$, denoted by $q_\theta$, is then given by the

pushforward density of $q_0$ under $T_\theta$, i.e.,

$$\log q_\theta(x) = \log q_0\left(T_\theta^{-1}(x)\right) + \log\left|\det\frac{\partial T_\theta^{-1}(x)}{\partial x}\right|, \tag{2}$$

see also Appendix A for general remarks on the notation. We focus on applications for which normalizing flows are trained to closely approximate a ground-truth target density $p(x) = \frac{1}{\mathcal{Z}}\exp(-E(x))$, where the energy $E : \mathbb{R}^d \to \mathbb{R}$ is known in closed-from but the partition function $\mathcal{Z} = \int_{\mathbb{R}^d} e^{-E(x)}\mathrm{d}x$ is intractable. To this end, there are two widely established training methods:

**Reverse KL training** relies on self-sampling from the flow and minimizes the reverse KL divergence

$$D_{\mathrm{KL}}(q_\theta, p) = \mathbb{E}_{x\sim q_\theta}\left[E(x) + \log q_\theta(x)\right] + \text{const.} \tag{3}$$

Since reverse KL training is based on self-sampling, the flow needs to be evaluated in the base-to-target direction $T_\theta$.

**Forward KL training** requires samples from the target density $p$ and is equivalent to maximum likelihood training

$$D_{\mathrm{KL}}(p, q_\theta) = \mathbb{E}_{x\sim p}\left[-\log q_\theta(x)\right] + \text{const.} \tag{4}$$

Since forward KL training requires the calculation of the density $q_\theta(x)$, the flow needs to be evaluated in the target-to-base direction $T_\theta^{-1}$, see (2).

As mentioned before, one typically uses a combined forward and reverse training to guide the self-sampling to high probability regions of the target density. When choosing a normalizing flow architecture for this task, it is therefore essential that both directions $T_\theta$ and $T_\theta^{-1}$ can be evaluated with reasonable efficiency. As a result, the following types of architectures are of practical relevance:

**Coupling Flows** are arguable the most widely used (see, e.g., Noé et al. (2019); Albergo et al. (2019); Nicoli et al. (2020); Matthews et al. (2022); Midgley et al. (2023); Huang et al. (2020)). They split the vector $x_l \in \mathbb{R}^d$ in two components

$$x_l = (x_l^{\mathrm{trans}}, x_l^{\mathrm{cond}}), \tag{5}$$

with $x_l^{\mathrm{trans}} \in \mathbb{R}^k$ and $x_l^{\mathrm{cond}} \in \mathbb{R}^{d-k}$ for $k \in \{1, \ldots, d-1\}$. The map $T_{l+1,\theta_{l+1}}$ is then given by

$$x_{l+1,i}^{\mathrm{trans}} = f_{\theta,i}(x_l^{\mathrm{trans}}, x_l^{\mathrm{cond}}) := \tau(x_{l,i}^{\mathrm{trans}}, h_{\theta,i}(x_l^{\mathrm{cond}})), \qquad \forall i \in \{1, \ldots, k\}, \tag{6a}$$

$$x_{l+1}^{\mathrm{cond}} = x_l^{\mathrm{cond}}, \tag{6b}$$

where $f_\theta : \mathbb{R}^k \times \mathbb{R}^{d-k} \to \mathbb{R}^k$, $\tau : \mathbb{R} \times \mathbb{R}^m \to \mathbb{R}$ are invertible maps with respect to their first argument for any choice of the second argument and $h_{\theta,i} : \mathbb{R}^{d-k} \to \mathbb{R}^m$ is the $i$-th output of a neural network. Note that the function $f_\theta$ acts on the components of $x_l^{\mathrm{trans}}$ element-wise.

There are broadly two types of coupling flows with different choices for the transformation $\tau$:

1. Explicitly invertible flows have the appealing property that the inverse map $T_{l+1,\theta_{l+1}}^{-1}$ can be calculated in closed-form and as efficiently as the forward map $T_{l+1,\theta_{l+1}}$. A particular example of this type of flows are affine coupling flows (Dinh et al., 2014; 2017) that use an affine transformation $\tau$, i.e.,

$$x_{l+1}^{\mathrm{trans}} = f_\theta(x_l^{\mathrm{trans}}, x_l^{\mathrm{cond}}) = \sigma_\theta(x_l^{\mathrm{cond}}) \odot x_l^{\mathrm{trans}} + \mu_\theta(x_l^{\mathrm{cond}}), \tag{7a}$$

$$x_{l+1}^{\mathrm{cond}} = x_l^{\mathrm{cond}}, \tag{7b}$$

with $h_\theta = (\sigma_\theta, \mu_\theta)$. Another example are neural spline flows (Durkan et al., 2019) which use splines instead of an affine transformation.

2. Implicitly invertible flows use a map $\tau$ whose inverse can only be obtained numerically, such as a mixture of non-compact projectors (Kanwar et al., 2020; Rezende et al., 2020) or smooth bump functions (Köhler et al., 2021). This often results in more expressive flows in particular in the context of normalizing flows on manifolds (Rezende et al., 2020). Recently, it has been shown in Köhler et al. (2021) that implicit differentiation can be used to train these types of flows using the forward KL objective.

**Continuous Normalizing Flows** use an ordinary differential equation (ODE) which relates to the bijection $T_\theta : \mathbb{R}^d \to \mathbb{R}^d$, allowing for straightforward implementation of equivariances, but typically coming with high computational costs (Chen et al., 2018).

Notably, autoregressive flows (Huang et al., 2018; Jaini et al., 2019) are less relevant in the context of learning a target density $p$ because they only permit fast evaluation in one direction and there is no training method based on implicit differentiation. As a result, they are not considered in this work.

## 3 PATH GRADIENTS FOR REVERSE KL

In this section, we introduce path gradients and show how they are related to the gradient of the reverse KL objective. The basic definition of path gradients is as follows:

**Definition 3.1.** The path gradient of a function $\varphi(\theta, T_\theta(x_0))$ is given by

$$\blacktriangledown_\theta \varphi(\theta, T_\theta(x_0)) := \left. \frac{\partial \varphi(\theta, x)}{\partial x} \right|_{x = T_\theta(x_0)} \frac{\partial T_\theta(x_0)}{\partial \theta} . \tag{8}$$

Note that the total derivative of the function $\varphi$ can be decomposed in the following way:

$$\frac{\mathrm{d}}{\mathrm{d}\theta} \varphi(\theta, T_\theta(x_0)) = \blacktriangledown_\theta \varphi(\theta, T_\theta(x_0)) + \left. \frac{\partial}{\partial \theta} \varphi(\theta, x) \right|_{x = T_\theta(x_0)} . \tag{9}$$

The path gradient therefore only takes the parameter dependence of the sampling path $T_\theta$ into account, but does not capture any explicit parameter dependence denoted by the second term. This decomposition was applied by Roeder et al. (2017) to the gradient of the reverse KL divergence to obtain the notable result

$$\frac{\mathrm{d}}{\mathrm{d}\theta} D_{\mathrm{KL}}(q_\theta, p) = \mathbb{E}_{x_0 \sim q_0} \left[ \blacktriangledown_\theta \left( E(T_\theta(x_0)) + \log q_\theta(T_\theta(x_0)) \right) \right], \tag{10}$$

where we have used the fact that $\mathbb{E}_{x_0 \sim q_0} \left[ \frac{\partial}{\partial \theta} \log q_\theta(T_\theta(x_0)) \right] = 0$. Thus, an unbiased estimator for the gradient of the KL divergence is given by

$$\mathcal{G}_{\mathrm{path}} := \frac{1}{N} \sum_{n=1}^{N} \blacktriangledown_\theta \left[ E(T_\theta(x_0^{(n)})) + \log q_\theta(T_\theta(x_0^{(n)})) \right], \tag{11}$$

where $x_0^{(n)} \sim q_0$ are i.i.d. samples. This path gradient estimator has been observed to have lower variance compared to the standard gradient estimator (Roeder et al., 2017; Tucker et al., 2019; Agrawal et al., 2020).

As the total derivative of the energy agrees with the path gradient of the energy function, i.e., $\frac{\mathrm{d}}{\mathrm{d}\theta} E(T_\theta(x_0)) = \blacktriangledown_\theta E(T_\theta(x_0))$, the first term in the estimator can be straightforwardly calculated using automatic differentiation. The second term, involving the path score $\blacktriangledown_\theta \log q_\theta(T_\theta(x_0))$, is however non-trivial as the path gradient through the sampling path $T_\theta$ has to be disentangled from the explicit parameter dependence in $q_\theta$. Recently, Vaitl et al. (2022a) proposed a method to calculate this term using the following steps:

1. Sample from the flow without building the computational graph:

$$x' = \text{stop\_gradients}(T_\theta(x_0)) \qquad \text{for} \quad x_0 \sim q_0 . \tag{12}$$

2. Calculate the gradient of the density with respect to the sample $x'$ using automatic differentiation:

$$G = \frac{\partial}{\partial x'} \log q_\theta(x') = \frac{\partial}{\partial x'} \left( \log q_0(T_\theta^{-1}(x')) + \log \det \left| \frac{\partial T_\theta^{-1}(x')}{\partial x'} \right| \right) . \tag{13}$$

3. Calculate the path gradient using a vector Jacobian product which can be efficiently calculated by standard reverse-mode automatic differentiation[*]:

$$\blacktriangledown_\theta \log q_\theta(T_\theta(x_0)) = G \frac{\partial T_\theta(x_0)}{\partial \theta} . \tag{14}$$

---

[*]Following standard convention in the autograd community, we adopt the convention that $G$ is a row vector. This is because the differential $\mathrm{d}f = \mathrm{d}x_i \frac{\partial f}{\partial x_i}$ of a function $f$ is a one-form and thus an element of the co-tangent space.

This method therefore requires the evaluation of both directions $T_\theta$ and $T_\theta^{-1}$. For implicitly invertible flows, backpropagation through a numerical inversion per training iteration is thus required, which is often prohibitively expensive.

Even in the best case scenario, i.e., for flows that can be evaluated in both directions with the same computational costs, such as RealNVP (Dinh et al., 2017), this algorithm has significant computational overhead. Specifically, it has roughly the costs of five forward passes: one for the sampling (12) and two each for the two gradient calculations (13) and (14) (which each require a forward as well as a backward pass). This is to be contrasted with the costs of the standard gradient estimator which only requires a single forward as well as a backward pass, i.e., has the cost of roughly two forward passes. In practical implementations, typically a runtime overhead of a factor of two instead of $\frac{5}{2}$ is observed for the path gradient estimator compared to the standard gradient estimator.

### 3.1 FAST PATH GRADIENT ESTIMATOR

In the following, we outline a fast method to estimate the path gradient. An important downside of the algorithm outlined in the last section is that one has to evaluate the flow in both directions $T_\theta$ and $T_\theta^{-1}$. The basic idea of the method outlined in the following is to calculate the derivative $\partial_x \log q_\theta(x)$ of the flow model recursively during sampling process. As a result, the flow only needs to be calculated in the forward direction $T_\theta$ as the second step in the path gradient algorithm discussed in the previous section can be avoided. In more detail, the calculation of the path gradient proceeds in two steps:

1. The sample $x = T_\theta(x_0)$ and the gradient $G = \frac{\partial}{\partial x} \log q_\theta(x)$ can be calculated alongside the sampling process using the recursive relation derived below.

2. The path gradient is then calculated with automatic differentiation using a vector Jacobian product, where, however, the forward pass $T_\theta(x_0)$ does not have to be recomputed:

$$\blacktriangledown_\theta \log q_\theta(T_\theta(x_0)) = G \frac{\partial T_\theta(x_0)}{\partial \theta} \,. \tag{15}$$

The recursion to calculate the derivative $\partial_x \log q_\theta(x)$ is as follows:

**Proposition 3.2** (Gradient recursion). *Using the diffeomorphism $T_l$, the derivative of the induced probability can be computed recursively as follows*

$$\frac{\partial \log q_{\theta,l+1}(x_{l+1})}{\partial x_{l+1}} = \frac{\partial \log q_{\theta,l}(x_l)}{\partial x_l} \left( \frac{\partial T_{l,\theta_l}(x_l)}{\partial x_l} \right)^{-1} - \frac{\partial \log \left| \det \frac{\partial T_{l,\theta_l}(x_l)}{\partial x_l} \right|}{\partial x_l} \left( \frac{\partial T_{l,\theta_l}(x_l)}{\partial x_l} \right)^{-1} . \tag{16}$$

For general $T_l$, computing the inverse Jacobian $\left( \partial T_{l,\theta_l}(x_l)/\partial x_l \right)^{-1}$ entails a time and space complexity higher than $\mathcal{O}(d)$, which is the complexity of the standard gradient estimator. For autoregressive flows, the total complexity is $\mathcal{O}(d^2)$, since its Jacobian is triangular. For coupling-type flows, we can simplify and speed up the recursion to have linear complexity in the number of dimensions, i.e. $\mathcal{O}(d)$. We state the recursive gradient computations for these kind of flows in the following proposition.

**Proposition 3.3** (Recursive gradient computations for coupling flows). *For a coupling flow,*

$$x_{l+1}^{\text{trans}} = f_\theta(x_l^{\text{trans}}, x_l^{\text{cond}}) \quad and \quad x_{l+1}^{\text{cond}} = x_l^{\text{cond}} , \tag{17}$$

*the derivative of the logarithmic density can be calculated recursively as follows*

$$\frac{\partial \log q_{\theta,l+1}(x_{l+1})}{\partial x_{l+1}^{\text{trans}}} = \frac{\partial \log q_{\theta,l}(x_l)}{\partial x_l^{\text{trans}}} \left( \frac{\partial f_\theta(x_l^{\text{trans}}, x_l^{\text{cond}})}{\partial x_l^{\text{trans}}} \right)^{-1}$$
$$- \frac{\partial}{\partial x_l^{\text{trans}}} \log \left| \det \frac{\partial f_\theta(x_l^{\text{trans}}, x_l^{\text{cond}})}{\partial x_l^{\text{trans}}} \right| \left( \frac{\partial f_\theta(x_l^{\text{trans}}, x_l^{\text{cond}})}{\partial x_l^{\text{trans}}} \right)^{-1} , \tag{18}$$

$$\frac{\partial \log q_{\theta,l+1}(x_{l+1})}{\partial x_{l+1}^{\text{cond}}} = \frac{\partial \log q_{\theta,l}(x_l)}{\partial x_l^{\text{cond}}} - \frac{\partial \log q_{\theta,l+1}(x_{l+1})}{\partial x_{l+1}^{\text{trans}}} \frac{\partial f_\theta(x_l^{\text{trans}}, x_l^{\text{cond}})}{\partial x_l^{\text{cond}}}$$
$$- \frac{\partial}{\partial x_l^{\text{cond}}} \log \left| \det \frac{\partial f_\theta(x_l^{\text{trans}}, x_l^{\text{cond}})}{\partial x_l^{\text{trans}}} \right| , \tag{19}$$

*starting with*

$$\frac{\partial \log q_{\theta,0}(x_0)}{\partial x_0^{\text{trans}}} = \frac{\partial \log q_0(x_0)}{\partial x_0^{\text{trans}}}, \qquad\qquad \frac{\partial \log q_{\theta,0}(x_0)}{\partial x_0^{\text{cond}}} = \frac{\partial \log q_0(x_0)}{\partial x_0^{\text{cond}}}. \qquad (20)$$

For a proof, see Appendix B.1. We stress that the Jacobian $\partial f_\theta(x_l^{\text{trans}}, x_l^{\text{cond}})/\partial x_l^{\text{trans}}$ is a $k \times k$ square and invertible matrix, since $f_\theta(\cdot, x_l^{\text{cond}})$ is bijective for any $x_l^{\text{cond}} \in \mathbb{R}^{d-k}$, see (6).

**Implicitly Invertible Flows.** An interesting property of the recursions in Proposition 3.3 is that they only involve (derivatives of) $f_\theta(x_l^{\text{trans}}, x_l^{\text{cond}})$ and can thus be evaluated during the sampling from the flow. As such, they are directly applicable to implicitly invertible flows. Further note that the Jacobian $\partial f_\theta(x_l^{\text{trans}}, x_l^{\text{cond}})/\partial x_l^{\text{trans}}$ can be inverted in linear time $\mathcal{O}(d)$, as it is a diagonal matrix; the function $f$ acts element-wise on $x_l^{\text{trans}}$, see (6). Therefore, the recursion has the decisive advantage that no numerical inversions need to be performed. In particular, there is no need for prohibitive backpropagation through such an inversion.

**Explicitly Invertible Flows.** For explicitly invertible normalizing flows — the most favorable setup for the baseline method from Vaitl et al. (2022a) — the runtime reduction appears to be more mild at first sight. The algorithm has roughly the cost of three forward passes: one each for the calculation of both $x$ and $G$ and one more for the backward pass when calculating the path gradient in (15). This is to be compared to the cost of five forward passes for the baseline method by Vaitl et al. (2022a) to calculate path gradients and two forward passes for the standard total gradient. However, this rough counting neglects the synergy between the sampling process $x = T(x_0)$ and the calculation of the score $G$. As we will show experimentally in Section 5, the actual runtime increase is only about forty percent compared to the standard total gradient.

Finally, let us note that for the aforementioned popular case of affine coupling flows our recursion from Proposition 3.3 takes a particular form. Since fewer terms need to be calculated, the following recursion gives an additional improvement in computational speed.

**Corollary 3.4** (Recursive gradient computations for affine coupling flows). *For an affine coupling flow (7), the recursion for the derivative of the logarithmic density can be simplified to*

$$\frac{\partial \log q_{\theta,l+1}(x_{l+1})}{\partial x_{l+1}^{\text{trans}}} = \frac{\partial \log q_{\theta,l}(x_l)}{\partial x_l^{\text{trans}}} \oslash \sigma_\theta(x_l^{\text{cond}}), \qquad (21)$$

$$\frac{\partial \log q_{\theta,l+1}(x_{l+1})}{\partial x_{l+1}^{\text{cond}}} = \frac{\partial \log q_{\theta,l}(x_l)}{\partial x_l^{\text{cond}}} - \frac{\partial \log q_{\theta,l+1}(x_{l+1})}{\partial x_{l+1}^{\text{trans}}} \left( \frac{\partial \sigma_\theta(x_l^{\text{cond}})}{\partial x_l^{\text{cond}}} \odot \overline{x}_l^{\text{trans}} + \frac{\partial \mu_\theta(x_l^{\text{cond}})}{\partial x_l^{\text{cond}}} \right)$$

$$- \frac{\partial}{\partial x_l^{\text{cond}}} \log \Big| \prod_{i=1}^{k} \sigma_{\theta,i}(x_l^{\text{cond}}) \Big|,$$

*where $\overline{x}_l^{\text{trans}}$ is a matrix with entries $\left(\overline{x}_l^{\text{trans}}\right)_{ij} := x_{l,i}^{\text{trans}}$ for $i \in \{1,\ldots,k\}, j \in \{1,\ldots,d-k\}$.*

For a proof, see Appendix B.2. Additionally, we show in Appendix C that the fast path gradient derived by Vaitl et al. (2022b) for continuous normalizing flows can be rederived using analogous steps as in Proposition 3.3. Our results therefore unify path gradient calculations of coupling flows with the analogous ones for continuous normalizing flows.

Finally, we note that a further distinctive strength of the proposed fast path gradient algorithm is that it can be performed at constant memory costs. Specifically, the calculation of $G$ can be done without saving any activations. Similarly, the activations needed for the vector Jacobian product (15) can be calculated alongside the backward pass as $T_\theta(x_0) = x$ is known using the techniques of Gomez et al. (2017).

## 4 PATH GRADIENTS FOR THE FORWARD KL DIVERGENCE

For training normalizing flows with the forward KL divergence, previous works have mainly relied on reweighting path gradients (Vaitl et al., 2022a). Specifically, their basic underlying trick is to rewrite the expectation value with respect to the ground-truth $p$ as an expectation value with respect to the model $q_\theta$

$$D_{\text{KL}}(p, q_\theta) = \mathbb{E}_{x \sim p}\left[\log \frac{p(x)}{q_\theta(x)}\right] = \mathbb{E}_{x \sim q_\theta}\left[\frac{p(x)}{q_\theta(x)} \log \frac{p(x)}{q_\theta(x)}\right]. \qquad (22)$$

For this reweighted loss, suitable path gradient estimators were then derived in Tucker et al. (2019). Reweighting, however, has the significant downside that it leads to estimators with prohibitive variance — especially for high-dimensional problems and in the early stages of training (Hartmann & Richter, 2021). As a result, the proposed estimators cannot be applied in a scalable fashion (Dhaka et al., 2021; Geffner & Domke, 2021a).

In the following, we will propose a general method to apply path gradients to forward KL training without the need for reweighting. To this end, we first notice that the forward KL of densities in data space can be equivalently rewritten as a reverse KL in base space, namely

$$D_{\mathrm{KL}}(p, q_\theta) = D_{\mathrm{KL}}(p_{\theta,0}, q_0) \,, \tag{23}$$

where we have defined the pullback of the target density $p$ to base space as follows

$$p_{\theta,0}(x_0) := p(T_\theta(x_0)) \left| \det \frac{\partial T_\theta(x_0)}{\partial x_0} \right| \,. \tag{24}$$

We refer to Papamakarios et al. (2021) for a proof. As a result, all results derived for the reverse KL case in the last sections also apply verbatim to the forward KL case if one exchanges:

$$q_\theta \longleftrightarrow p_{\theta,0} \,, \qquad p \longleftrightarrow q_0 \,, \qquad x_0 \longleftrightarrow x \,, \qquad T_\theta(x_0) \longleftrightarrow T_\theta^{-1}(x) \,. \tag{25}$$

In particular, the fast path gradient estimators can be straightforwardly applied. More precisely, the following statement holds:

**Proposition 4.1** (Path gradient for forward KL). *For the derivative of the forward KL divergence $D_{\mathrm{KL}}(p|q_\theta)$ w.r.t. the parameter $\theta$ it holds*

$$\frac{\mathrm{d}}{\mathrm{d}\theta} D_{\mathrm{KL}}(p|q_\theta) = \mathbb{E}_{x \sim p}\left[ \blacktriangledown_\theta \log \frac{p_{\theta,0}}{q_0}(T_\theta^{-1}(x)) \right], \tag{26}$$

*where $p_{\theta,0}(x_0) := p(T_\theta(x_0)) \left| \det \frac{\partial T_\theta(x_0)}{\partial x_0} \right|$ is the pullback of the target density $p$ to base space.*

For a proof, see Appendix B.3. Note that if $p$ is only known in unnormalized form, so is its pullback $p_{\theta,0}$. However, this has no impact on the derived result as it only involves derivatives of the log density for which the normalization is irrelevant. The following comments are in order:

- The proposed path gradient for maximum likelihood training provides an attractive mechanism to incorporate the known closed-form target energy function into the training process. In particular, this can help to alleviate overfitting, cf. Figures 1 and 8 — a particularly relevant concern as the forward training often uses a low amount of samples which entails the risk of density collapse on the individual samples for standard maximum likelihood training. The information about the energy function helps the path-gradient-based training to avoid this undesired behaviour. On the other hand, forward KL path gradient training cannot be used if the target energy function is not known such as in image generation tasks.

- As for path gradients of the reverse KL, we expect lower variance of the Monte Carlo estimator of (26) compared to standard maximum likelihood gradient estimators. In particular, we note that at the optimum $q_\theta = p$ the variance of the gradient estimator vanishes.

- The proof in Appendix B shows that the so-called generalized doubly reparameterized gradient proposed in Bauer & Mnih (2021) in the context of hierarchical VAEs can be brought in the the same form as the path gradient for the forward KL objective derived in this section. However, only our formulation elucidates the symmetry between the forward and reverse objective and therefore allows the application for fast path gradient estimators.

## 5 NUMERICAL EXPERIMENTS

In this section, we compare our fast path gradients with the conventional approaches for several normalizing flow architectures, both using forward and reverse KL optimization. We consider target densities with applications in machine learning (Gaussian mixture model) as well as physics ($U(1)$ gauge theory, and $\phi^4$ lattice model). We refer to Appendix E for further details. *

---

* Code for reproducing the experiments for GMM and $U(1)$ at github.com/lenz3000/unified-path-gradients.

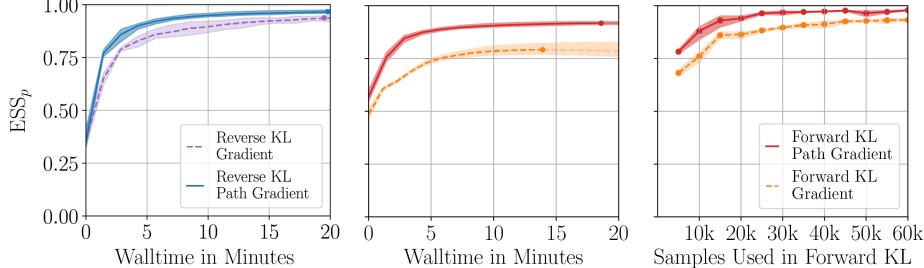

Figure 1: Effective sample size (ESS) over the training iterations for a Gaussian mixture model using the forward and the reverse KL divergence. The intervals denote the standard error over 5 runs. The best performance is indicated by a dot with subsequent faded average performance in the left and center figure. For the forward KL, we compare multiple hyperparameter settings (see Appendix E) and plot the respective best runs in the central plot. The right plot displays a stereotypical dependency on the data set size for fixed hyperparameters, see Tables 3, 4 and 5 for more details. We can see that, typically, path gradients perform better than standard maximum likelihood gradients.

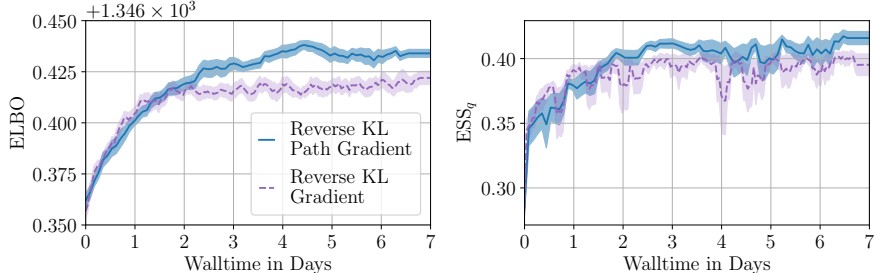

Figure 2: Training the $U(1)$ flow for Lattice Gauge Theory. Shaded area shows standard error over 4 runs. The Reverse KL Path Gradients reach higher performance and exhibit less erratic behavior.

**Gaussian Mixture Model.** As a tractable multimodal example, we consider a Gaussian mixture model in $\mathbb{R}^d$ with $\sigma^2 = 0.5$, i.e. we choose the energy function

$$E(x) = -\log \sum_{\mu \in \{-1,1\}^d} \mathcal{N}(x; \mu, \sigma^2 \, \mathrm{I}_d) \tag{27}$$

Note that the number of modes of the corresponding target density increases exponentially in the dimensions, i.e. we have $2^d$ modes in total. We choose $d = 6$, resulting in $64$ modes. As shown in Figure 1, for most choices of hyperparameters, path gradient training outperforms the standard training objectives. In Figures 5 to 7 in the appendix we present further experiments, showing that path gradient estimators are indeed often better and never significantly worse than standard estimators. The slight overhead in runtime is therefore more than compensated by better training convergence. The additional information about the ground-truth energy function included in the forward path gradient training alleviates overfitting in forward KL training, see the discussion in Section 4.

$\phi^4$ **Field Theory** can be described by a random vector $\phi \in \mathbb{R}^d$, whose entries $\phi_u$ represent the values of the corresponding field across a $16 \times 8$ lattice. The lattice positions are encoded in the set $\Lambda \subset \mathbb{N}^2$. We assume periodic boundary conditions of the lattice. The random vector $\phi$ admits the density[*] $p(\phi) = \frac{1}{\mathcal{Z}} \exp(-S(\phi))$ with action

$$S(\phi) = \sum_{u,v \in \Lambda} \phi_u \triangle_{uv} \phi_v + \sum_{u \in \Lambda} \left( m^2 \phi_u^2 + \lambda \phi_u^4 \right), \tag{28}$$

where $\triangle_{uv}$ is the lattice Laplacian. The parameters $m$ and $\lambda$ are the bare mass and coupling, respectively. We choose the value of these parameters such that they lie in the so-called critical region, as this is the most challenging regime. We refer to Gattringer & Lang (2009) for more details

---

[*]Note that, by slightly abusing notation, $\phi$ plays the role of what was $x$ before.

Table 1: Results of the experiments from Section 5. We measure the approximation quality of the variational density $q_\theta$ by the effective sampling size (ESS) plus standard deviations, where higher is better, i.e., $100\%$ indicates perfect approximation, see Appendix E for details.

| | | Reverse KL | | Forward KL | |
|---|---|---|---|---|---|
| | | Gradient | Path Gradient | Gradient | Path Gradient |
| GMM | $\mathrm{ESS}_p$ | $92.2 \pm 0.0$ | $\mathbf{97.4 \pm 0.0}$ | $79.1 \pm 0.0$ | $\mathbf{91.8 \pm 0.0}$ |
| | $\mathrm{ESS}_q$ | $93.0 \pm 0.0$ | $\mathbf{97.4 \pm 0.0}$ | $84.1 \pm 0.0$ | $\mathbf{91.8 \pm 0.0}$ |
| $\phi^4$ | $\mathrm{ESS}_p$ | $85.6 \pm 0.1$ | $\mathbf{96.0 \pm 0.1}$ | $85.1 \pm 0.1$ | $\mathbf{95.6 \pm 0.0}$ |
| | $\mathrm{ESS}_q$ | $85.6 \pm 0.1$ | $\mathbf{96.0 \pm 0.1}$ | $85.1 \pm 0.1$ | $\mathbf{95.6 \pm 0.0}$ |
| $U(1)$ | $\mathrm{ESS}_q$ | $40.1 \pm 0.0$ | $\mathbf{41.1 \pm 0.0}$ | — | — |
| | ELBO | $1346.42 \pm .01$ | $\mathbf{1346.43 \pm .00}$ | — | — |

Table 2: Factor of runtime increase (mean and standard deviation) in comparison to the standard gradient estimator, i.e., runtime path gradient/runtime standard gradient on an A100-80GB GPU. The upper set of experiments cover the explicitly invertible flows, applied to $\phi^4$ as treated in the experiments. The lower set covers implicitly invertible flows applied to $U(1)$ theory.

| | Algorithm | Runtime factor with batch size | | |
|---|---|---|---|---|
| | | 64 | 1024 | 8192 |
| Expl | Alg. 1 **(ours)** | $\mathbf{1.6 \pm 0.1}$ | $\mathbf{1.4 \pm 0.1}$ | $\mathbf{1.4 \pm 0.0}$ |
| | Alg. 2 (Vaitl et al., 2022a) | $2.1 \pm 0.1$ | $2.2 \pm 0.1$ | $2.1 \pm 0.0$ |
| Impl | Alg. 1 **(ours)** | $\mathbf{2.2 \pm 0.0}$ | $\mathbf{2.0 \pm 0.1}$ | $\mathbf{2.3 \pm 0.0}$ |
| | Alg. 2 + Köhler et al. (2021) | $17.5 \pm 0.2$ | $11.0 \pm 0.1$ | $8.2 \pm 0.0$ |

on the underlying physics. Training is performed using both the forward and reverse KL objective with and without path gradients. For the flow, the same affine-coupling-based architecture as in Nicoli et al. (2020) is used. Samples for forward KL and ESS are generated using Hybrid Monte Carlo. We refer to Appendix E for more details. The path gradient training again outperforms the standard objective for both forward and reverse training, see Table 1.

**Gauge Theory** was recently widely studied in the context of normalizing flows (Kanwar et al., 2020; Albergo et al., 2021; Finkenrath, 2022; Bacchio et al., 2023; Cranmer et al., 2023) as it provides an ideal setting for illustrating the power of inductive biases. This is because the theory's action has a gauge symmetry, i.e., a symmetry which acts with independent group elements for each lattice site, see Gattringer & Lang (2009) for more details. Crucially, the field takes values in the circle group $U(1)$. Thus, flows on manifolds need to be considered. We use the flow architecture proposed by Kanwar et al. (2020) which is only implicitly invertible. Sampling from the ground-truth distribution with Hybrid Monte Carlo is very challenging due to critical slowing down and we therefore refrain from forward KL training and forward ESS evaluation. Table 1 and Figure 2 demonstrate that path gradients lead to overall better approximation quality.

**Runtime Comparison.** In Table 2, we compare the runtime of our method to relevant baselines both for the ex- and implicitly invertible flows. To obtain a strong baseline for the latter, we use implicit differentiation as in Köhler et al. (2021) to avoid costly backpropagation through the numerical inversion. Our method is significantly faster than the baselines. We refer to Appendix E for a detailed analysis of how this runtime comparison scales with the chosen accuracy of the numerical inversion. Briefly summarized, we find that our method compares favorable to the baseline irrespective of the chosen accuracy.

## 6 CONCLUSION

We have introduced a fast and unified method to estimate path gradients for normalizing flows which can be applied to both forward and reverse training. We find that the path gradient training consistently improves training for both the reverse and forward case. An appealing property of path-gradient maximum likelihood is that it can take information about the ground truth energy function into account and thereby acts as a particularly natural form of regularization. Our fast path gradient estimators are several multiples faster than the previous state-of-the-art, they are applicable accross a broad range of NF architectures, and considerably narrow the runtime gap to the standard gradient while preserving the desirable variance reduction.

**Acknowledgements.** L.V. thanks Matteo Gätzner for his preliminary work on GDReGs. L.W. thanks Jason Rinnert for visualization help and acknowledges support by the Federal Ministry of Education and Research (BMBF) for BIFOLD (01IS18037A). The research of L.R. has been partially funded by Deutsche Forschungsgemeinschaft (DFG) through the grant CRC 1114 "Scaling Cascades in Complex Systems" (project A05, project number 235221301). P.K. wants to thank Andreas Loukas for useful discussions.

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

## APPENDIX

## A  NOTATION

For a function $\psi : \mathbb{R}^d \to \mathbb{R}$ we denote with $\frac{\mathrm{d}\psi}{\mathrm{d}x}$ its total derivative, for a function $\varphi : \mathbb{R}^d \times \mathbb{R}^p \to \mathbb{R}$ we denote with $\frac{\partial \varphi(x,y)}{\partial x}$ its partial derivative w.r.t. the first argument $x \in \mathbb{R}^d$ and for a function $T : \mathbb{R}^d \to \mathbb{R}^d$ we denote by $\frac{\partial T}{\partial x}$ its Jacobian. For ease of notation, we sometimes use the shorthand notation $\frac{\partial x_l}{\partial x_{l+1}}$ for $\frac{\partial}{\partial x_{l+1}} T_{l+1,\theta_{l+1}}^{-1}(x_{l+1})$, keeping in mind that $x_{l+1} = T_{l+1,\theta_{l+1}}(x_l)$. We write $\blacktriangledown\varphi$ for the path gradient of $\varphi$ as defined in Definition 3.1. The symbol $\odot$ denotes elementwise multiplication and $\oslash$ denotes elementwise division.

## B  PROOFS

In this section we collect the proofs of our propositions and corollaries, which we will recall here for convenience.

### B.1  RECURSIVE GRADIENT COMPUTATIONS FOR COUPLING FLOWS

First, let us recall the following proposition from Section 3.1.

**Proposition 3.2** (Gradient recursion). *Using the diffeomorphism $T_l$, the derivative of the induced probability can be computed recursively as follows*

$$\frac{\partial \log q_{\theta,l+1}(x_{l+1})}{\partial x_{l+1}} = \frac{\partial \log q_{\theta,l}(x_l)}{\partial x_l} \left( \frac{\partial T_{l,\theta_l}(x_l)}{\partial x_l} \right)^{-1} - \frac{\partial \log \left| \det \frac{\partial T_{l,\theta_l}(x_l)}{\partial x_l} \right|}{\partial x_l} \left( \frac{\partial T_{l,\theta_l}(x_l)}{\partial x_l} \right)^{-1}. \tag{16}$$

*Proof.* The basic definition of flows consisting of compositions

$$\log q_{\theta,l+1}(x_{l+1}) = \log q_{\theta,l}(T_{l,\theta_l}^{-1}(x_{l+1})) + \log \left| \det \frac{\partial T_{l,\theta_l}^{-1}(x_{l+1})}{\partial x_{l+1}} \right| \tag{29}$$

implies the following recursion

$$\frac{\partial \log q_{\theta,l+1}(x_{l+1})}{\partial x_{l+1}} = \frac{\partial \log q_{\theta,l}(T_{l,\theta_l}^{-1}(x_{l+1}))}{\partial x_{l+1}} + \frac{\partial}{\partial x_{l+1}} \log \left| \det \frac{\partial T_{l,\theta_l}^{-1}(x_{l+1})}{\partial x_{l+1}} \right|. \tag{30}$$

Since for general normalizing flows, the inverse $T_l^{-1}(x_{l+1})$ is not efficiently computable, we apply the chain rule and the inverse function theorem

$$\frac{\partial \log q_{\theta,l+1}(x_{l+1})}{\partial x_{l+1}} = \frac{\partial \log q_{\theta,l}(x_l)}{\partial x_l} \frac{\partial T_{l,\theta_l}^{-1}(x_{l+1})}{\partial x_{l+1}} - \frac{\partial}{\partial x_l} \left( \log \left| \det \frac{\partial T_{l,\theta_l}(x_l)}{\partial x_l} \right| \right) \frac{\partial T_{l,\theta_l}^{-1}(x_{l+1})}{\partial x_{l+1}} \tag{31a}$$

$$= \frac{\partial \log q_{\theta,l}(x_l)}{\partial x_l} \left( \frac{\partial T_{l,\theta_l}(x_l)}{\partial x_l} \right)^{-1} - \frac{\partial}{\partial x_l} \left( \log \left| \det \frac{\partial T_{l,\theta_l}(x_l)}{\partial x_l} \right| \right) \left( \frac{\partial T_{l,\theta_l}(x_l)}{\partial x_l} \right)^{-1}. \tag{31b}$$

$$\square$$

We can break down the recursive operation further in the next proposition.

**Proposition 3.3** (Recursive gradient computations for coupling flows). *For a coupling flow,*

$$x_{l+1}^{\mathrm{trans}} = f_\theta(x_l^{\mathrm{trans}}, x_l^{\mathrm{cond}}) \quad and \quad x_{l+1}^{\mathrm{cond}} = x_l^{\mathrm{cond}}, \tag{17}$$

*the derivative of the logarithmic density can be calculated recursively as follows*

$$\frac{\partial \log q_{\theta,l+1}(x_{l+1})}{\partial x_{l+1}^{\text{trans}}} = \frac{\partial \log q_{\theta,l}(x_l)}{\partial x_l^{\text{trans}}} \left( \frac{\partial f_\theta(x_l^{\text{trans}}, x_l^{\text{cond}})}{\partial x_l^{\text{trans}}} \right)^{-1}$$
$$- \frac{\partial}{\partial x_l^{\text{trans}}} \log \left| \det \frac{\partial f_\theta(x_l^{\text{trans}}, x_l^{\text{cond}})}{\partial x_l^{\text{trans}}} \right| \left( \frac{\partial f_\theta(x_l^{\text{trans}}, x_l^{\text{cond}})}{\partial x_l^{\text{trans}}} \right)^{-1}, \quad (18)$$

$$\frac{\partial \log q_{\theta,l+1}(x_{l+1})}{\partial x_{l+1}^{\text{cond}}} = \frac{\partial \log q_{\theta,l}(x_l)}{\partial x_l^{\text{cond}}} - \frac{\partial \log q_{\theta,l+1}(x_{l+1})}{\partial x_{l+1}^{\text{trans}}} \frac{\partial f_\theta(x_l^{\text{trans}}, x_l^{\text{cond}})}{\partial x_l^{\text{cond}}}$$
$$- \frac{\partial}{\partial x_l^{\text{cond}}} \log \left| \det \frac{\partial f_\theta(x_l^{\text{trans}}, x_l^{\text{cond}})}{\partial x_l^{\text{trans}}} \right|, \quad (19)$$

*starting with*

$$\frac{\partial \log q_{\theta,0}(x_0)}{\partial x_0^{\text{trans}}} = \frac{\partial \log q_0(x_0)}{\partial x_0^{\text{trans}}}, \qquad \frac{\partial \log q_{\theta,0}(x_0)}{\partial x_0^{\text{cond}}} = \frac{\partial \log q_0(x_0)}{\partial x_0^{\text{cond}}}. \quad (20)$$

*Proof.* For coupling flows $x_l = (x_l^{\text{trans}}, x_l^{\text{cond}})$ and $x_{l+1}^{\text{cond}} = x_l^{\text{cond}}$. This implies that the determinant of the Jacobian is

$$\det \frac{\partial x_l}{\partial x_{l+1}} = \det \begin{pmatrix} \frac{\partial x_l^{\text{trans}}}{\partial x_{l+1}^{\text{trans}}} & \frac{\partial x_l^{\text{trans}}}{\partial x_{l+1}^{\text{cond}}} \\ 0 & \mathbb{1} \end{pmatrix} = \det \frac{\partial x_l^{\text{trans}}}{\partial x_{l+1}^{\text{trans}}}. \quad (32)$$

Using this result and splitting into the transformed and conditional components, the recursion can be rewritten as follows

$$\frac{\partial \log q_{\theta,l+1}(x_{l+1})}{\partial x_{l+1}^{\text{trans}}} = \frac{\partial \log q_{\theta,l}(x_l)}{\partial x_{l+1}^{\text{trans}}} + \frac{\partial}{\partial x_{l+1}^{\text{trans}}} \log \left| \det \frac{\partial x_l^{\text{trans}}}{\partial x_{l+1}^{\text{trans}}} \right|, \quad (33)$$

$$\frac{\partial \log q_{\theta,l+1}(x_{l+1})}{\partial x_{l+1}^{\text{cond}}} = \frac{\partial \log q_{\theta,l}(x_l)}{\partial x_{l+1}^{\text{cond}}} + \frac{\partial}{\partial x_{l+1}^{\text{cond}}} \log \left| \det \frac{\partial x_l^{\text{trans}}}{\partial x_{l+1}^{\text{trans}}} \right|, \quad (34)$$

We want to rewrite the right-hand side of this recursion in terms of quantities that involve derivatives with respect to $x_l$. To this end, let us study derivatives w.r.t. $x_{l+1}^{\text{trans}}$ and $x_{l+1}^{\text{cond}}$, respectively.

For a generic function $\varphi : \mathbb{R}^k \times \mathbb{R}^{d-k} \to \mathbb{R}$ it holds via the chain rule that

$$\frac{\partial \varphi(x_l^{\text{trans}}, x_l^{\text{cond}})}{\partial x_{l+1}^{\text{trans}}} = \frac{\partial \varphi(x_l^{\text{trans}}, x_l^{\text{cond}})}{\partial x_l^{\text{trans}}} \frac{\partial x_l^{\text{trans}}}{\partial x_{l+1}^{\text{trans}}} + \frac{\partial \varphi(x_l^{\text{trans}}, x_l^{\text{cond}})}{\partial x_l^{\text{cond}}} \frac{\partial x_l^{\text{cond}}}{\partial x_{l+1}^{\text{trans}}}. \quad (35)$$

Noting that $x_l^{\text{trans}} = f_\theta^{-1}(x_{l+1}^{\text{trans}}, x_{l+1}^{\text{cond}})$ and noting that $x_{l+1}^{\text{cond}} = x_l^{\text{cond}}$, we can compute

$$\frac{\partial \varphi(x_l^{\text{trans}}, x_l^{\text{cond}})}{\partial x_{l+1}^{\text{trans}}} = \frac{\partial \varphi(x_l^{\text{trans}}, x_l^{\text{cond}})}{\partial x_l^{\text{trans}}} \frac{\partial f_\theta^{-1}(x_{l+1}^{\text{trans}}, x_{l+1}^{\text{cond}})}{\partial x_{l+1}^{\text{trans}}} \quad (36a)$$

$$= \frac{\partial \varphi(x_l^{\text{trans}}, x_l^{\text{cond}})}{\partial x_l^{\text{trans}}} \left( \frac{\partial f_\theta(x_l^{\text{trans}}, x_l^{\text{cond}})}{\partial x_l^{\text{trans}}} \right)^{-1}, \quad (36b)$$

where we used the fact that the Jacobian of the inverse is the inverse of the Jacobian due to the inverse function theorem.

Similarly, we can write

$$\frac{\partial \varphi(x_l^{\text{trans}}, x_l^{\text{cond}})}{\partial x_{l+1}^{\text{cond}}} = \frac{\partial \varphi(x_l^{\text{trans}}, x_l^{\text{cond}})}{\partial x_l^{\text{trans}}} \frac{\partial x_l^{\text{trans}}}{\partial x_{l+1}^{\text{cond}}} + \frac{\partial \varphi(x_l^{\text{trans}}, x_l^{\text{cond}})}{\partial x_l^{\text{cond}}} \frac{\partial x_l^{\text{cond}}}{\partial x_{l+1}^{\text{cond}}} \quad (37a)$$

$$= \frac{\partial \varphi(x_l^{\text{trans}}, x_l^{\text{cond}})}{\partial x_l^{\text{trans}}} \frac{\partial x_l^{\text{trans}}}{\partial x_{l+1}^{\text{cond}}} + \frac{\partial \varphi(x_l^{\text{trans}}, x_l^{\text{cond}})}{\partial x_l^{\text{cond}}}, \quad (37b)$$

where we have used the decomposition (6) of a coupling flow.

Note that the Jacobian $\frac{\partial x_l^{\text{trans}}}{\partial x_{l+1}^{\text{cond}}}$ is not necessarily invertible and may not even be a square matrix. We have assumed that $x_l^{\text{trans}} = f_\theta^{-1}(x_{l+1}^{\text{trans}}, x_{l+1}^{\text{cond}})$ is invertible only with respect to its first argument (for any choice of its last). So while the Jacobians $\frac{\partial x_l^{\text{trans}}}{\partial x_{l+1}^{\text{trans}}}$ and $\frac{\partial f_\theta(x_l^{\text{trans}}, x_l^{\text{cond}})}{\partial x_l^{\text{trans}}}$ are square and invertible matrices, the same cannot be said for the Jacobian $\frac{\partial x_l^{\text{trans}}}{\partial x_{l+1}^{\text{cond}}}$. However, we can use the following trick

$$0 = \frac{\partial x_{l+1}^{\text{trans}}}{\partial x_{l+1}^{\text{cond}}} = \frac{\partial f_\theta(x_l^{\text{trans}}, x_l^{\text{cond}})}{\partial x_{l+1}^{\text{cond}}} \tag{38a}$$

$$= \frac{\partial f_\theta(x_l^{\text{trans}}, x_l^{\text{cond}})}{\partial x_l^{\text{trans}}} \frac{\partial x_l^{\text{trans}}}{\partial x_{l+1}^{\text{cond}}} + \frac{\partial f_\theta(x_l^{\text{trans}}, x_l^{\text{cond}})}{\partial x_l^{\text{cond}}} \frac{\partial x_l^{\text{cond}}}{\partial x_{l+1}^{\text{cond}}} \tag{38b}$$

$$= \frac{\partial f_\theta(x_l^{\text{trans}}, x_l^{\text{cond}})}{\partial x_l^{\text{trans}}} \frac{\partial x_l^{\text{trans}}}{\partial x_{l+1}^{\text{cond}}} + \frac{\partial f_\theta(x_l^{\text{trans}}, x_l^{\text{cond}})}{\partial x_l^{\text{cond}}}. \tag{38c}$$

Since the Jacobian $\frac{\partial f_\theta(x_l^{\text{trans}}, x_l^{\text{cond}})}{\partial x_l^{\text{trans}}}$ is invertible, the above statement is equivalent to

$$\frac{\partial x_l^{\text{trans}}}{\partial x_{l+1}^{\text{cond}}} = -\left( \frac{\partial f_\theta(x_l^{\text{trans}}, x_l^{\text{cond}})}{\partial x_l^{\text{trans}}} \right)^{-1} \frac{\partial f_\theta(x_l^{\text{trans}}, x_l^{\text{cond}})}{\partial x_l^{\text{cond}}}. \tag{39}$$

Substituting this result into (37b) yields

$$\begin{aligned}
\frac{\partial \varphi(x_l^{\text{trans}}, x_l^{\text{cond}})}{\partial x_{l+1}^{\text{cond}}} &= -\frac{\partial \varphi(x_l^{\text{trans}}, x_l^{\text{cond}})}{\partial x_l^{\text{trans}}} \left( \frac{\partial f_\theta(x_l^{\text{trans}}, x_l^{\text{cond}})}{\partial x_l^{\text{trans}}} \right)^{-1} \frac{\partial f_\theta(x_l^{\text{trans}}, x_l^{\text{cond}})}{\partial x_l^{\text{cond}}} \\
&\quad + \frac{\partial \varphi(x_l^{\text{trans}}, x_l^{\text{cond}})}{\partial x_l^{\text{cond}}}.
\end{aligned} \tag{40}$$

Next, we note that the determinant of the Jacobian can be rewritten as

$$\log \left| \det \frac{\partial f_\theta^{-1}(x_{l+1}^{\text{trans}}, x_{l+1}^{\text{cond}})}{\partial x_{l+1}^{\text{trans}}} \right| = -\log \left| \det \frac{\partial f_\theta(x_l^{\text{trans}}, x_l^{\text{cond}})}{\partial x_l^{\text{trans}}} \right|, \tag{41}$$

using again the inverse function theorem.

Now, plugging (36), (40) and (41) into (33) for suitable choices of $\varphi$, we can rewrite the recursion in the desired form:

$$\begin{aligned}
\frac{\partial \log q_\theta(x_{l+1})}{\partial x_{l+1}^{\text{trans}}} &= \frac{\partial \log q_\theta(x_l)}{\partial x_l^{\text{trans}}} \left( \frac{\partial f_\theta(x_l^{\text{trans}}, x_l^{\text{cond}})}{\partial x_l^{\text{trans}}} \right)^{-1} \\
&\quad - \frac{\partial}{\partial x_l^{\text{trans}}} \log \left| \det \frac{\partial f_\theta(x_l^{\text{trans}}, x_l^{\text{cond}})}{\partial x_l^{\text{trans}}} \right| \left( \frac{\partial f_\theta(x_l^{\text{trans}}, x_l^{\text{cond}})}{\partial x_l^{\text{trans}}} \right)^{-1}.
\end{aligned} \tag{42}$$

This is precisely the form stated in the proposition. Analogously, plugging (36), (40) and (41) into (34), the conditional component can be rewritten as

$$\begin{aligned}
\frac{\partial \log q_\theta(x_{l+1})}{\partial x_{l+1}^{\text{cond}}} &= \frac{\partial \log q_\theta(x_l)}{\partial x_l^{\text{cond}}} - \frac{\partial \log q_\theta(x_l)}{\partial x_l^{\text{trans}}} \left( \frac{\partial f_\theta(x_l^{\text{trans}}, x_l^{\text{cond}})}{\partial x_l^{\text{trans}}} \right)^{-1} \frac{\partial f_\theta(x_l^{\text{trans}}, x_l^{\text{cond}})}{\partial x_l^{\text{cond}}} \\
&\quad + \frac{\partial}{\partial x_l^{\text{trans}}} \log \left| \det \frac{\partial f_\theta(x_l^{\text{trans}}, x_l^{\text{cond}})}{\partial x_l^{\text{trans}}} \right| \left( \frac{\partial f_\theta(x_l^{\text{trans}}, x_l^{\text{cond}})}{\partial x_l^{\text{trans}}} \right)^{-1} \frac{\partial f_\theta(x_l^{\text{trans}}, x_l^{\text{cond}})}{\partial x_l^{\text{cond}}} \\
&\quad - \frac{\partial}{\partial x_l^{\text{cond}}} \log \left| \det \frac{\partial f_\theta(x_l^{\text{trans}}, x_l^{\text{cond}})}{\partial x_l^{\text{trans}}} \right|.
\end{aligned} \tag{43}$$

This can be brought in more compact form by noticing that the term $\frac{\partial \log q_\theta(x_{l+1})}{\partial x_{l+1}^{\text{trans}}}$ appears in the expression for the conditional component. Using this, we can rewrite the conditional component as

$$
\begin{aligned}
\frac{\partial \log q_\theta(x_{l+1})}{\partial x_{l+1}^{\text{cond}}} =& \frac{\partial \log q_\theta(x_l)}{\partial x_l^{\text{cond}}} - \frac{\partial \log q_\theta(x_{l+1})}{\partial x_{l+1}^{\text{trans}}} \frac{\partial f_\theta(x_l^{\text{trans}}, x_l^{\text{cond}})}{\partial x_l^{\text{cond}}} \\
& - \frac{\partial}{\partial x_l^{\text{cond}}} \log \left| \det \frac{\partial f_\theta(x_l^{\text{trans}}, x_l^{\text{cond}})}{\partial x_l^{\text{trans}}} \right|,
\end{aligned}
\tag{44}
$$

which is precisely the form stated in the proposition. □

### B.2 RECURSIVE GRADIENT COMPUTATIONS FOR AFFINE COUPLING FLOWS

Let us recall the following corollary from Section 3.1.

**Corollary 3.4** (Recursive gradient computations for affine coupling flows). *For an affine coupling flow (7), the recursion for the derivative of the logarithmic density can be simplified to*

$$
\frac{\partial \log q_{\theta,l+1}(x_{l+1})}{\partial x_{l+1}^{\text{trans}}} = \frac{\partial \log q_{\theta,l}(x_l)}{\partial x_l^{\text{trans}}} \oslash \sigma_\theta(x_l^{\text{cond}}),
\tag{21}
$$

$$
\begin{aligned}
\frac{\partial \log q_{\theta,l+1}(x_{l+1})}{\partial x_{l+1}^{\text{cond}}} =& \frac{\partial \log q_{\theta,l}(x_l)}{\partial x_l^{\text{cond}}} - \frac{\partial \log q_{\theta,l+1}(x_{l+1})}{\partial x_{l+1}^{\text{trans}}} \left( \frac{\partial \sigma_\theta(x_l^{\text{cond}})}{\partial x_l^{\text{cond}}} \odot \overline{x}_l^{\text{trans}} + \frac{\partial \mu_\theta(x_l^{\text{cond}})}{\partial x_l^{\text{cond}}} \right) \\
& - \frac{\partial}{\partial x_l^{\text{cond}}} \log \left| \prod_{i=1}^{k} \sigma_{\theta,i}(x_l^{\text{cond}}) \right|,
\end{aligned}
$$

*where $\overline{x}_l^{\text{trans}}$ is a matrix with entries $\left(\overline{x}_l^{\text{trans}}\right)_{ij} := x_{l,i}^{\text{trans}}$ for $i \in \{1, \ldots, k\}, j \in \{1, \ldots, d-k\}$.*

*Proof.* Affine coupling flows are defined by

$$
x_{l+1}^{\text{trans}} = f_\theta(x_l^{\text{trans}}, x_l^{\text{cond}}) = \sigma_\theta(x_l^{\text{cond}}) \odot x_l^{\text{trans}} + \mu_\theta(x_l^{\text{cond}}), \qquad x_{l+1}^{\text{cond}} = x_l^{\text{cond}}.
\tag{45}
$$

This implies for the Jacobian

$$
\frac{\partial f_\theta(x_l^{\text{trans}}, x_l^{\text{cond}})}{\partial x_l^{\text{trans}}} = \text{diag}\left(\sigma_\theta(x_l^{\text{cond}})\right),
\tag{46}
$$

and for its determinant it holds

$$
\det \frac{\partial f_\theta(x_l^{\text{trans}}, x_l^{\text{cond}})}{\partial x_l^{\text{trans}}} = \prod_{i=1}^{k} \sigma_{\theta,i}\left(x_l^{\text{cond}}\right),
\tag{47}
$$

which notably does not depend on $x_l^{\text{trans}}$. Therefore, the recursion for the transformed component simplifies to

$$
\begin{aligned}
\frac{\partial \log q_{\theta,l+1}(x_{l+1})}{\partial x_{l+1}^{\text{trans}}} =& \frac{\partial \log q_{\theta,l}(x_l)}{\partial x_l^{\text{trans}}} \left( \frac{\partial f_\theta(x_l^{\text{trans}}, x_l^{\text{cond}})}{\partial x_l^{\text{trans}}} \right)^{-1} \\
& - \frac{\partial}{\partial x_l^{\text{trans}}} \left( \frac{\partial f_\theta(x_l^{\text{trans}}, x_l^{\text{cond}})}{\partial x_l^{\text{trans}}} \right)^{-1} \log \left| \det \frac{\partial f_\theta(x_l^{\text{trans}}, x_l^{\text{cond}})}{\partial x_l^{\text{trans}}} \right| \quad \text{(48a)} \\
=& \frac{\partial \log q_{\theta,l}(x_l)}{\partial x_l^{\text{trans}}} \oslash \sigma_\theta(x_l^{\text{cond}}), \quad \text{(48b)}
\end{aligned}
$$

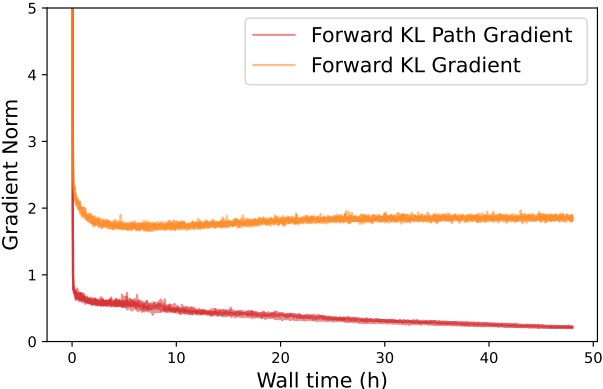

Figure 3: Gradient norm during training of the $\phi^4$-experiments. The norm of the path gradient estimator is closer to zero than the norm of the standard gradient estimator when the target density is well approximated, indicating lower variance.

which is precisely the form stated in the proposition. The recursion for the conditional component simplifies to

$$
\frac{\partial \log q_{\theta,l+1}(x_{l+1})}{\partial x_{l+1}^{\mathrm{cond}}} = \frac{\partial \log q_{\theta,l}(x_l)}{\partial x_l^{\mathrm{cond}}} - \frac{\partial \log q_{\theta,l+1}(x_{l+1})}{\partial x_{l+1}^{\mathrm{trans}}} \frac{\partial f_\theta(x_l^{\mathrm{trans}}, x_l^{\mathrm{cond}})}{\partial x_l^{\mathrm{cond}}} \tag{49a}
$$

$$
- \frac{\partial}{\partial x_l^{\mathrm{cond}}} \log \left| \det \frac{\partial f_\theta(x_l^{\mathrm{trans}}, x_l^{\mathrm{cond}})}{\partial x_l^{\mathrm{trans}}} \right|
$$

$$
= \frac{\partial \log q_{\theta,l}(x_l)}{\partial x_l^{\mathrm{cond}}} - \frac{\partial \log q_{\theta,l+1}(x_{l+1})}{\partial x_{l+1}^{\mathrm{trans}}} \left( \frac{\partial \sigma_\theta(x_l^{\mathrm{cond}})}{\partial x_l^{\mathrm{cond}}} \odot \overline{x}_l^{\mathrm{trans}} + \frac{\partial \mu_\theta(x_l^{\mathrm{cond}})}{\partial x_l^{\mathrm{cond}}} \right)
$$

$$
- \frac{\partial}{\partial x_l^{\mathrm{cond}}} \log \left| \prod_{i=1}^{k} \sigma_{\theta,i}(x_l^{\mathrm{cond}}) \right|, \tag{49b}
$$

where $\overline{x}_l^{\mathrm{trans}}$ is a matrix with entries $\left( \overline{x}_l^{\mathrm{trans}} \right)_{ij} := x_{l,i}^{\mathrm{trans}}$ for $i \in \{1, \ldots, k\}, j \in \{1, \ldots, d-k\}$. This shows the claim.

$\square$

### B.3 PATH GRADIENTS FOR THE FORWARD KL DIVERGENCE

In this section, we prove Proposition 4.1 and lay out its implications for path gradient estimators, both for the forward and reverse KL divergence. In particular, we will discuss the favorable variance properties of path gradients in the case of the forward KL divergence, which we also verify experimentally in Figure 3.

**Proposition 4.1** (Path gradient for forward KL). *For the derivative of the forward KL divergence $D_{\mathrm{KL}}(p|q_\theta)$ w.r.t. the parameter $\theta$ it holds*

$$
\frac{\mathrm{d}}{\mathrm{d}\theta} D_{\mathrm{KL}}(p|q_\theta) = \mathbb{E}_{x \sim p} \left[ \blacktriangledown_\theta \log \frac{p_{\theta,0}}{q_0}(T_\theta^{-1}(x)) \right], \tag{26}
$$

*where $p_{\theta,0}(x_0) := p(T_\theta(x_0)) \left| \det \frac{\partial T_\theta(x_0)}{\partial x_0} \right|$ is the pullback of the target density $p$ to base space.*

*Proof.* Let us first note that

$$D_{\mathrm{KL}}(p|q_\theta) = \mathbb{E}_{x \sim p} \left[ \log p(x) - \log \left| \det \frac{\partial T_\theta^{-1}(x)}{\partial x} \right| - \log q_0(T_\theta^{-1}(x)) \right] \tag{50a}$$

$$= \mathbb{E}_{x_0 \sim p_{\theta,0}} \left[ \log p(T_\theta(x_0)) + \log \left| \det \frac{\partial T_\theta(x_0)}{\partial x_0} \right| - \log q_0(x_0) \right] \tag{50b}$$

$$= D_{\mathrm{KL}}(p_{\theta,0}|q_0). \tag{50c}$$

Note that in (50) we have essentially transformed a forward into a reverse KL divergence. By looking at the problem of minimizing the forward KL divergence $D_{\mathrm{KL}}(p|q_\theta)$ — from target density $p$ to the variational density $q_\theta$ — as a reverse KL divergence $D_{\mathrm{KL}}(p_{\theta,0}|q_0)$ — from a variational density $p_{\theta,0}$ to target density $q_0$ —, we can employ the tools that exist for optimizing the reverse KL divergence. In particular, we can now apply the standard path gradients as derived by Roeder et al. (2017) and Tucker et al. (2019). We compute

$$\frac{\mathrm{d}}{\mathrm{d}\theta} D_{\mathrm{KL}}(p_{\theta,0}|q_0) = \mathbb{E}_{x \sim p} \left[ \frac{\partial}{\partial \theta} \log \frac{p_{\theta,0}}{q_0}(T_\theta^{-1}(x)) \right] \tag{51a}$$

$$= \mathbb{E}_{x \sim p} \left[ \frac{\partial}{\partial x_0} \left( \log \frac{p_{\theta,0}}{q_0}(x_0) \right) \frac{\partial T_\theta^{-1}(x)}{\partial \theta} + \frac{\partial \log p_{\theta,0}(x_0)}{\partial \theta} \bigg|_{x_0 = T_\theta^{-1}(x)} \right] \tag{51b}$$

$$= \mathbb{E}_{x \sim p} \left[ \blacktriangledown_\theta \log \frac{p_{\theta,0}}{q_0}(T_\theta^{-1}(x)) \right] + \mathbb{E}_{x_0 \sim p_{\theta,0}} \left[ \frac{\partial \log p_{\theta,0}(x_0)}{\partial \theta} \right] \tag{51c}$$

$$= \mathbb{E}_{x \sim p} \left[ \blacktriangledown_\theta \log \frac{p_{\theta,0}}{q_0}(T_\theta^{-1}(x)) \right], \tag{51d}$$

where we used the fact that the score $\frac{\partial}{\partial \theta} \log p_{\theta,0}(x_0)$ vanishes in expectation over $p_{\theta,0}$, since

$$\mathbb{E}_{x_0 \sim p_{\theta,0}} \left[ \frac{\partial}{\partial \theta} \log p_{\theta,0}(x_0) \right] = \int_{\mathbb{R}^d} \frac{\partial}{\partial \theta} \log p_{\theta,0}(x_0) p_{\theta,0}(x_0) \mathrm{d}x_0 \tag{52a}$$

$$= \int_{\mathbb{R}^d} \frac{\partial}{\partial \theta} p_{\theta,0}(x_0) \mathrm{d}x_0 \tag{52b}$$

$$= \frac{\partial}{\partial \theta} \int_{\mathbb{R}^d} p_{\theta,0}(x_0) \mathrm{d}x_0 \tag{52c}$$

$$= \frac{\partial}{\partial \theta} 1 = 0. \tag{52d}$$

Now, the statement follows by combining (50) and (51). $\square$

### B.3.1 Variance: Sticking the Landing Property

The covariance matrix of the score term is known to be the Fisher Information (cf., e.g., Vaitl et al. (2022a)). In this case, it is the Fisher Information $\mathcal{I}_0(\theta)$ of the pullback density $p_{\theta,0}$, defined as

$$\mathcal{I}_0(\theta) := \mathbb{E}_{x_0 \sim p_{\theta,0}} \left[ \frac{\partial}{\partial \theta} \log p_{\theta,0}(x_0)^\top \frac{\partial}{\partial \theta} \log p_{\theta,0}(x_0) \right]. \tag{53}$$

If the model perfectly approximates the target density, i.e. $p_{\theta,0}(x_0) = q_0(x_0)$ for all $x_0 \in \mathbb{R}^d$, then the path gradient term

$$\blacktriangledown_\theta \left( \log \frac{p_{\theta,0}}{q_0}(T_\theta^{-1}(x)) \right) = \underbrace{\frac{\partial}{\partial x_0} \left( \log p_{\theta,0}(x_0) - \log q_0(x_0) \right)}_{=0} \bigg|_{x_0 = T_\theta^{-1}(x)} \frac{\partial T_\theta^{-1}(x)}{\partial \theta} \equiv 0$$

is zero almost surely. This implies that the path gradient estimator has zero variance in the limit of perfect approximation (which is sometimes called *sticking the landing*), while the standard gradient estimator has non-vanishing covariance $\mathcal{I}_0(\theta)/N$, where $N$ is the sample size of the Monte Carlo estimator. Note that these results are exactly analogous to the reverse KL path gradients, noting the relations in (25). Experimentally, we find for the forward KL divergence that the gradient norm indeed behaves as in previous works (Roeder et al., 2017; Vaitl et al., 2022a), i.e. it exhibits the vanishing gradient properties. This is illustrated in Figure 3.

### B.3.2 MOTIVATION FOR REGULARIZATION

The favorable behavior of path gradients for the forward KL divergence, which we have defined in Proposition 4.1, can potentially be understood by identifying the additional terms appearing in the gradient as having a regularizing effect. To this end, let us compare the standard maximum likelihood gradients with the path gradients. The former is given by

$$\frac{\mathrm{d}}{\mathrm{d}\theta} D_{\mathrm{KL}}(p|q_\theta) = -\mathbb{E}_{x\sim p}\left[\frac{\mathrm{d}}{\mathrm{d}\theta}\log q_\theta(x)\right] \tag{54a}$$

$$= -\mathbb{E}_{x\sim p}\left[\frac{\mathrm{d}}{\mathrm{d}\theta}\left(\log q_0(T_\theta^{-1}(x)) + \log\left|\det\frac{\partial T_\theta^{-1}(x)}{\partial x}\right|\right)\right], \tag{54b}$$

whereas the latter can be computed as

$$\frac{\mathrm{d}}{\mathrm{d}\theta} D_{\mathrm{KL}}(p|q_\theta) = \mathbb{E}_{x\sim p}\left[\blacktriangledown_\theta\left(\log\frac{q_0}{p_{\theta,0}}(T_\theta^{-1}(x))\right)\right] \tag{55a}$$

$$= \mathbb{E}_{x\sim p}\left[\frac{\partial}{\partial x_0}\left(\log q_0(x_0)) - \log\left|\det\frac{\partial T_\theta(x_0)}{\partial x_0}\right|\right.\right.$$
$$\left.\left. - \log p(T_\theta(x_0))\right)\right|_{x_0=T^{-1}(x)}\frac{\partial T_\theta^{-1}(x)}{\partial\theta}\right]. \tag{55b}$$

Note that only in the path gradient version (55), the target density $p$ appears and we conjecture that incorporating this information helps to not overfit to the given data sample. Crucially, in many applications, $p$ is (up to normalization) given explicitly such that (55) can indeed be readily computed. Because of the duality of the KL divergence, the regularization property also appears for the reverse KL, where the term involving $q_0$ does not appear in the standard gradient estimator, whereas for the path gradients it does.

### B.3.3 RELATION TO GDREG

The above gradient formula (26) can be seen as a special case of the Generalized Doubly-Reparameterized Gradient Estimator (GDReG) derived in Bauer & Mnih (2021), by noting that

$$\mathbb{E}_{x\sim p}\left[\blacktriangledown_\theta\log\frac{p_{\theta,0}}{q_0}(T_\theta^{-1}(x))\right] = -\mathbb{E}_{x\sim p}\left[\blacktriangledown_\theta\log\frac{p}{q_\theta}(T_\theta(x_0))\Big|_{x_0=T_\theta^{-1}(x)}\right]. \tag{56}$$

This can be seen as follows:

$$\blacktriangledown_\theta\log\frac{p}{q_\theta}(T_\theta(z))\Big|_{x_0=T_\theta^{-1}(x)} \tag{57a}$$

$$= \frac{\partial\left(\log p(x) - \log\det\left|\frac{\partial T_\theta^{-1}(x)}{\partial x}\right| - \log q_0(T_\theta^{-1}(x))\right)}{\partial x}\frac{\partial T_\theta(x_0)}{\partial\theta}\Big|_{x_0=T_\theta^{-1}(x)} \tag{57b}$$

$$= -\frac{\partial\left(\log p(x) - \log\det\left|\frac{\partial T_\theta^{-1}(x)}{\partial x}\right| - \log q_0(T_\theta^{-1}(x))\right)}{\partial x}\frac{\partial T_\theta(x_0)}{\partial x_0}\frac{\partial T_\theta^{-1}(x)}{\partial\theta} \tag{57c}$$

$$= -\frac{\partial\left(\log p(T_\theta(x_0)) + \log\det\left|\frac{\partial T_\theta(x_0)}{\partial x_0}\right| - \log q_0(x_0)\right)}{\partial x_0}\frac{\partial T_\theta^{-1}(x)}{\partial\theta} \tag{57d}$$

$$= -\blacktriangledown_\theta\log\frac{p_{\theta,0}}{q_0}(T_\theta^{-1}(x)), \tag{57e}$$

where in (57c) we have used the identity

$$0 = \frac{\partial T_\theta(T_\theta^{-1}(x))}{\partial\theta} = \frac{\partial T_\theta(z)}{\partial z}\frac{\partial T_\theta^{-1}(x)}{\partial\theta} + \frac{\partial T_\theta(z)}{\partial\theta}\Big|_{x_0=T_\theta^{-1}(x)}. \tag{58}$$

However, only our derivation allows to interpret the estimator as an instance of the standard sticking-the-landing trick of eliminating a score function, which then motivates variance reduced estimators and allows us to use the proposed Algorithm 1. Further, the time for computing (26) is expected to be significantly lower than for the GDReG estimator since in this form we can employ our proposed algorithm for efficiently computing the path gradient.

# C   RELATION TO FAST PATH GRADIENT FOR CONTINUOUS NORMALIZING FLOWS

In this section, we demonstrate that the recently proposed fast path gradient for continuous normalizing flows (CNFs) proposed by Vaitl et al. (2022b) can be obtained using completely analogous reasoning as in our derivation for the fast path gradient of the coupling flows.

We first note that the algorithm described in Section 3.1 can be applied verbatim to the CNF case with

$$X_t = \mathcal{T}_\theta(X_0) := X_0 + \int_0^t v_\theta(X_s, s)\, \mathrm{d}s\,, \tag{59}$$

where $v_\theta : \mathbb{R}^d \times \mathbb{R} \to \mathbb{R}^d$ is the generating vector field of the CNF. Note that $T_\theta$ defined in (1) can be interpreted as a discretization of the flow $\mathcal{T}_\theta$ and $x$ in (1) can be seen as a discrete approximation of $X_t$. The only difference between the CNF and coupling case arises in the recursive relation to obtain the derivative

$$\frac{\partial \log q_\theta(X_t)}{\partial X_t}\,. \tag{60}$$

Vaitl et al. (2022b) proposed an ODE that can be evolved along with the sampling process. As we will demonstrate subsequently, this ODE can easily be recovered by using the same reasoning as we applied for the coupling flows.

We start from the observation that the ODE can be discretized as

$$x_{l+1} = x_l + v_\theta(x_l, l\Delta t)\Delta t\,. \tag{61}$$

for $l \in \{0, \ldots, \frac{t}{\Delta t} - 1\}$ in the sense that $X_{l\Delta t} = x_l + \mathcal{O}(\Delta t^2)$ . Here, we assume for convenience that the time increment $\Delta t > 0$ is chosen such that $\frac{t}{\Delta t}$ is an integer.

As for the coupling flows, we start from equation (31b), which for a discretized CNF is given by

$$\frac{\partial \log q_\theta(x_{l+1})}{\partial x_{l+1}} = \frac{\partial \log q_\theta(x_l)}{\partial x_{l+1}} + \frac{\partial}{\partial x_{l+1}} \log \left| \det \frac{\partial x_l}{\partial x_{l+1}} \right|\,. \tag{62}$$

Using the chain rule, this can be rewritten as

$$\frac{\partial \log q_\theta(x_{l+1})}{\partial x_{l+1}} = \frac{\partial \log q_\theta(x_l)}{\partial x_l} \frac{\partial x_l}{\partial x_{l+1}} + \left( \frac{\partial}{\partial x_l} \log \left| \det \frac{\partial x_l}{\partial x_{l+1}} \right| \right) \frac{\partial x_l}{\partial x_{l+1}} \tag{63a}$$

$$= \frac{\partial \log q_\theta(x_l)}{\partial x_l} \left( \frac{\partial x_{l+1}}{\partial x_l} \right)^{-1} + \left( \frac{\partial}{\partial x_l} \log \left| \det \frac{\partial x_l}{\partial x_{l+1}} \right| \right) \left( \frac{\partial x_{l+1}}{\partial x_l} \right)^{-1}\,. \tag{63b}$$

$$= \frac{\partial \log q_\theta(x_l)}{\partial x_l} \left( \frac{\partial x_{l+1}}{\partial x_l} \right)^{-1} - \left( \frac{\partial}{\partial x_l} \log \left| \det \frac{\partial x_{l+1}}{\partial x_l} \right| \right) \left( \frac{\partial x_{l+1}}{\partial x_l} \right)^{-1}\,. \tag{63c}$$

From the discretized ODE (61), it follows that

$$\left( \frac{\partial x_{l+1}}{\partial x_l} \right)^{-1} = \left( \mathbb{1} + \Delta t \frac{\partial v_\theta}{\partial x_l}(x_l, l\Delta t) \right)^{-1} = \mathbb{1} - \Delta t \frac{\partial v_\theta}{\partial x_l}(x_l, l\Delta t)\,. \tag{64}$$

Using this expression, we obtain

$$\frac{\partial \log q_\theta(x_{l+1})}{\partial x_{l+1}} = \left( \frac{\partial \log q_\theta(x_l)}{\partial x_l} - \frac{\partial}{\partial x_l} \log \left| \det \frac{\partial x_{l+1}}{\partial x_l} \right| \right) \left( \mathbb{1} - \Delta t \frac{\partial v_\theta}{\partial x_l}(x_l, l\Delta t) \right)\,. \tag{65}$$

Using the standard series expansion of the matrix-valued logarithm, we can check that

$$\log \left| \det \left( \mathbb{1} + \Delta t \frac{\partial v_\theta}{\partial x_l}(x_l, l\Delta t) \right) \right| = \sum_{n=1}^\infty \frac{(-1)^{k+1}}{k} \mathrm{Tr} \left( \frac{\partial v_\theta}{\partial x_l} \right)^k \Delta t^k = \Delta t\, \mathrm{Tr} \left[ \frac{\partial v_\theta}{\partial x_l}(x_l, l\Delta t) \right] + \mathcal{O}(\Delta t^2)\,, \tag{66}$$

where in the second equality, we have used that $\log \det A = \mathrm{Tr} \log A$ and $\log(1 + B) = \sum_{k=1}^{\infty} \frac{(-1)^{k+1}}{k} B^k$ for matrices $A$ and $B$. Substituting this expression, in the expression above, we obtain

$$\frac{1}{\Delta t} \left( \frac{\partial \log q_\theta(x_{l+1})}{\partial x_{l+1}} - \frac{\partial \log q_\theta(x_l)}{\partial x_l} \right) = -\frac{\partial \log q_\theta(x_l)}{\partial x_l} \frac{\partial v_\theta}{\partial x_l}(x_l, l\Delta t) - \frac{\partial}{\partial x_l}\mathrm{Tr}\left[\frac{\partial v_\theta}{\partial x_l}\right](x_l, l\Delta t) + \mathcal{O}(\Delta t).$$

Taking the limit $\Delta t \to 0$, we obtain precisely the ODE derived by Vaitl et al. (2022b):

$$\frac{\mathrm{d}}{\mathrm{d}t}\frac{\partial \log q_\theta(X_t)}{\partial X_t} = -\frac{\partial \log q_\theta(X_t)}{\partial X_t}\frac{\partial v_\theta}{\partial X_t}(X_t, t) - \frac{\partial}{\partial X_t}\mathrm{Tr}\left[\frac{\partial v_\theta}{\partial X_t}\right](X_t, t). \tag{67}$$

As a result, the fast path gradient derived in this manuscript unifies path gradient calculations of coupling flows with the analogous ones for CNFs.

## D    Algorithms

In this section we state the different algorithms used in our work. As discussed in Section 4, due to the duality of the KL divergence, we can employ the algorithms for both the forward and the reverse KL.

The different algorithms treated in this paper are:

1. The novel fast path gradient algorithm — shown in Algorithm 1.

2. As a baseline, the method proposed in Vaitl et al. (2022a) — shown in Algorithm 2.

3. As a further baseline, the same algorithm, amended to the GDReG estimator (Bauer & Mnih, 2021) — shown in Algorithm 3. We stress that this algorithm is an original proposal of this paper, however, we did not introduce it in detail in the main text as it is slower than our fast path gradient and therefore more of a side-product serving as a strong baseline.

The algorithms do not contain the path gradients for the target density, since those can be done readily. A graphical visualisation of the respective terms of the algorithms is provided in Figure 4.

---

**Algorithm 1:** Fast Path Gradient: computation of $\blacktriangledown_\theta \log q_\theta(T_\theta(x_0)), x_0 \sim q_0$

**Input:** base sample $x_0 \sim q_0$
**for** $l$ in $\{0, \dots, L-1\}$:                                                          ▷ joint forward pass and recursive equations
    Apply $T_{l+1, \theta_{l+1}}$ to compute $x_{l+1}$
    Compute $\frac{\partial \log q_{\theta, l+1}(x_{l+1})}{\partial x_{l+1}}$ according to Proposition 3.3
    **return** $\frac{\partial \log q_{\theta, L}(x_L)}{\partial x_L}\frac{\partial}{\partial \theta}x_L$                                          ▷ compute vector-Jacobian products

---

**Algorithm 2:** Path gradient: computation of $\blacktriangledown_\theta \log q_\theta(T_\theta(x_0)), x_0 \sim q_0$

**Input:** base sample $x_0 \sim q_0$
    $x' \leftarrow \mathrm{stop\_gradient}(T_\theta(x_0))$                                ▷ forward pass of $x_0$ through the flow without gradients
    $q_\theta(x') \leftarrow q_0(T_\theta^{-1}(x'))\left|\det \frac{\partial T_\theta^{-1}(x')}{\partial x'}\right|$                          ▷ reverse pass to calculate density
    $G \leftarrow \frac{\partial \log(q_\theta(x'))}{\partial x'}$                                                      ▷ compute gradient with respect to $x'$
    $x \leftarrow T_\theta(x_0)$                                                              ▷ standard forward pass
    **return** $G\frac{\partial}{\partial \theta}x$                                                   ▷ compute vector-Jacobian products

---

## E    Computational Details

In this section we elaborate on computational details.

**Algorithm 3:** Path gradient: computation of $\blacktriangledown_\theta \log q_\theta(T_\theta(x_0))|_{x_0=T_\theta^{-1}(x)}, x \sim p$

**Input:** target sample $x \sim p$

$q_\theta(x') \leftarrow q_0(T_\theta^{-1}(x')) \left| \det \frac{\partial T_\theta^{-1}(x')}{\partial x'} \right|$           $\triangleright$ reverse pass to calculate density

$G \leftarrow \frac{\partial \log(q_\theta(x'))}{\partial x'}$           $\triangleright$ compute gradient with respect to $x'$

$x_0' \leftarrow$ stop_gradient($T_\theta^{-1}(x)$)           $\triangleright$ copy $T_\theta^{-1}(x)$

$x \leftarrow T_\theta(x_0')$           $\triangleright$ standard forward pass

**return** $G \frac{\partial}{\partial \theta} x$           $\triangleright$ compute vector-Jacobian products

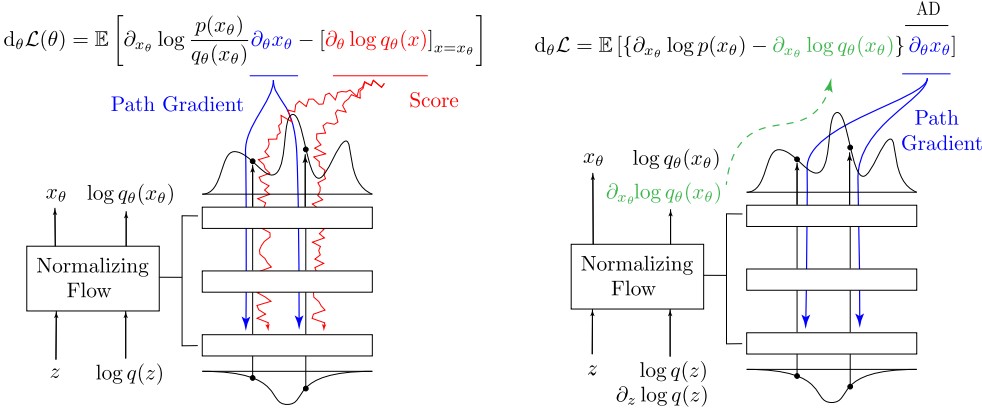

$$d_\theta \mathcal{L}(\theta) = \mathbb{E}\left[ \partial_{x_\theta} \log \frac{p(x_\theta)}{q_\theta(x_\theta)} \partial_\theta x_\theta - [\partial_\theta \log q_\theta(x)]_{x=x_\theta} \right]$$

$$d_\theta \mathcal{L} = \mathbb{E}\left[ \{\partial_{x_\theta} \log p(x_\theta) - \partial_{x_\theta} \log q_\theta(x_\theta)\} \overset{AD}{\partial_\theta x_\theta} \right]$$

Path Gradients and Score           Fast Path Gradients without Score

Figure 4: The gradient of the loss function $d_\theta \mathcal{L}(\theta)$ consists of the path gradient (blue) and a score term (red) — the latter vanishes in expectation, but has non-vanishing variance. The path gradient framework computes the necessary quantities to perform stochastic gradient descent with only the path gradients, eliminating the impact of the score term which tends to increase the variance of the gradient leading to suboptimal gradient estimation. We propose a fast algorithm for computing the Path Gradient estimator which evaluates $\partial_{x_\theta} \log q_\theta(x_\theta)$ (green) alongside the forward pass. The term $\partial_{x_\theta} \log p(x_\theta)$ is assumed to be available for a given energy function within the problem formulation. We can then compute the derivative of the log ratio, $\partial_{x_\theta} \log \left( p(x_\theta)/q_\theta(x_\theta) \right)$, which can be interpreted as a *scaling* of the gradient $\partial_\theta x_\theta$, the gradient with respect to the parameters. Thus the gradient $d_\theta \mathcal{L}(\theta)$ only consist of the path gradient, eliminating the negative influence of the score term.

**Evaluation Metrics.** As common, we assess the approximation quality of the variational density $q_\theta$ by the effective sampling size (ESS), defined as

$$\text{ESS} := \frac{1}{\mathbb{E}_{x \sim q_\theta}[w^2(x)]} = \frac{1}{\mathbb{E}_{x \sim p}[w(x)]}, \tag{68}$$

where $w(x) := \frac{p(x)}{q_\theta(x)}$ are the importance weights.

As can be seen in (68), the ESS can be computed with samples from either $q_\theta$ or $p$ (if available), which leads to two different estimators, namely

$$\widehat{\text{ESS}}_q = \frac{N}{\sum_{n=1}^N \widehat{w}_q^2(x^{(n)})}, \quad x^{(n)} \sim q_\theta, \qquad \widehat{\text{ESS}}_p = \frac{N}{\sum_{n=1}^N \widehat{w}_p(x^{(n)})}, \quad x^{(n)} \sim p, \tag{69}$$

where the normalization constant $Z$ appearing in the importance weights $w$ can as well be approximated either with samples from $q_\theta$ or $p$, respectively, namely

$$\widehat{w}_q(x) := \frac{e^{-S(x)}}{q_\theta(x)\widehat{Z}_q}, \qquad \widehat{Z}_q = \frac{1}{N}\sum_{n=1}^N \frac{e^{-S(x^{(n)})}}{q_\theta(x^{(n)})}, \qquad x^{(n)} \sim q_\theta, \tag{70}$$

or

$$\widehat{w}_p(x) := \frac{e^{-S(x)}}{q_\theta(x)\widehat{Z}_p}, \qquad \widehat{Z}_p = \left( \frac{1}{N} \sum_{n=1}^{N} \frac{q_\theta(x^{(n)})}{e^{-S(x^{(n)})}} \right)^{-1}, \qquad x^{(n)} \sim p. \qquad (71)$$

Note that $\widehat{\mathrm{ESS}}_q$ might be biased due to a potential mode collapse of $q_\theta$, which is not the case for $\widehat{\mathrm{ESS}}_p$. The ESS is a measure of how efficiently one can sample from the target distribution. E.g., an ESS of $0.5$ means that if we have $N$ samples from the sampling distribution $q_\theta$, the variance of the reweighted estimator is large as an estimator using $0.5N$ samples from the target distribution. A maximum ESS of $1$ occurs when the importance weights $\widehat{w}$ are exactly $1$ for every sample, the minimum ESS is $0$.

**Gradient Estimators.** As baselines for the path gradients we used the Maximum Likelihood gradient estimator as the Forward KL Standard Gradient estimator

$$\frac{\mathrm{d}}{\mathrm{d}\theta} D_{\mathrm{KL}}(p|q_\theta) \approx -\frac{1}{N} \sum_{n=1}^{N} \frac{\partial}{\partial\theta} \log q_\theta(x^{(n)}), \qquad x^{(n)} \sim p, \qquad (72)$$

and for the reverse KL we used the reparametrization trick gradients

$$\frac{\mathrm{d}}{\mathrm{d}\theta} D_{\mathrm{KL}}(q_\theta|p) \approx \frac{1}{N} \sum_{n=1}^{N} \frac{\mathrm{d}}{\mathrm{d}\theta} \left( \log \frac{q_\theta}{p}(T_\theta(x_0^{(n)})) \right), \qquad x_0^{(n)} \sim q_0. \qquad (73)$$

Note that both gradients are independent of the normalization constants of both $q_\theta$ and $p$.

**Gaussian Mixture Model.** We use a RealNVP Dinh et al. (2017) flow with weight normalization. The RealNVP flow contains 1,000 hidden neurons per layer and six coupling layers, each of which consists of 6 hidden neural networks layers with Tanh activation to generate the affine coupling parameters. We draw 10,000 samples from the Gaussian mixture model (GMM) for the forward KL training, thus mimicking a finite yet large sample set.

The superior performance of path gradients for Gaussian Mixture Models can be observed in Figure 1. In case of the reverse KL in the left plot, both path gradients and standard gradients achieve the same forward ESS, yet the path gradients converge faster in wall time, despite their increased computational cost of $40\%$ compared to the standard gradients.

The middle figure shows the improved performance of the forward KL path gradients compared to the standard non-path gradient. The forward KL path gradients converge faster to a high forward ESS and are able to maintain the achieved sampling efficiency compared to the standard gradients which decrease to a forward ESS of zero.

Finally, the right plot examines the different performance of forward KL path gradients in more detail by increasing the number of training samples on which the forward KL is optimized. The advantage of path gradients becomes evident even more as they generally surpass standard gradients earlier and achieve higher ultimate forward effective sampling sizes even for very large forward KL training sets.

Besides the forward ESS, the corresponding negative Loglikelihood (NLL) for both gradients and path gradients is plotted in Figure 9 for an increasing number of training samples. One observes how the NLL for the training and test data set differ only slightly compared to the performance gap of standard gradients measured in terms of NLL. Path gradients are robust even in low data regimes where standard gradients diverge strongly in terms of their training and test performance, indicating a tendency to overfit.

The results in the experiments in Figures 1 and 9 are obtained after $10,000$ optimization steps, with a learning rate of $0.00001$ with the Adam optimizer (Kingma & Ba, 2015) with a batch size of $4,000$. The target distribution $p$ is identical to the setup in Section 5 and the forward ESS and NLL are evaluated with $10,000$ test samples.

Finally, Figure 8 illustrates the performance gap between standard gradients and path gradients on a simple two dimensional multivariate Normal distribution. The visualization hints at a stronger

regularization for path gradients which are able to incorporate gradient information of the underlying ground truth energy function.

Figures 5, 6 and 7 show the $\text{ESS}_p$ over the course of the optimization for varying numbers of linear layers per coupling block, number of hidden neurons per linear layer and the batch size used for optimization trained with Forward KL Path Gradients and non-path Forward KL Gradients. While standard Forward KL gradients can *relatively* surpass Forward KL Path Gradients mildly in a few cases for smaller models, a larger model trained with Path Gradients doubles the mean of the $\text{ESS}_p$ in a direct comparison and improves upon the best Forward KL Gradients in *absolute* terms. Ultimately, the best performance as measured with the $\text{ESS}_p$ is achieved with larger models trained with path gradients. Importantly, path gradients provide increased robustness against overfitting as can be seen in the performance during training. Standard gradients for larger models tend to deteriorate their $\text{ESS}_p$.

The Tables 3, 4 and 5 collect the best $\text{ESS}_p$ achieved during optimization for the same combination of number of linear layers, number of hidden neurons per linear layer and batch size. This corresponds to saving checkpoints during training and choosing the respective model with the best $\text{ESS}_p$. The Forward KL Path Gradients outperform their non-path counterpart except for a few instances, which perform worse than larger models as measured in $\text{ESS}_p$ and trained with path gradients.

Table 3: We summarize the highest achieved average $ESS_p$ over the entire optimization which corresponds to the best attainable performance in terms of $\text{ESS}_p$ possible for that model capacity. The flow consists of 6 coupling layers, each with 2 linear layers, each of which has an increasing width (number of neurons).

| Layer Width | 10 | | 50 | | 100 | | 250 | | 500 | | 1000 | |
|---|---|---|---|---|---|---|---|---|---|---|---|---|
| Batch Size | Std | Path | Std | Path | Std | Path | Std | Path | Std | Path | Std | Path |
| 10 | 55.7 | **56.9** | 58.5 | **58.8** | 58.5 | **58.8** | 60.0 | **62.4** | 71.3 | **72.7** | **72.1** | 71.7 |
| 50 | 58.0 | **58.3** | 58.9 | **59.2** | 59.3 | **60.0** | 69.8 | **75.2** | 76.0 | **77.2** | 66.6 | **80.5** |
| 100 | 58.3 | **58.6** | 59.1 | **59.4** | 60.2 | **62.1** | 73.4 | **79.7** | 75.1 | **78.3** | 66.3 | **84.1** |
| 200 | 58.5 | **58.7** | 59.4 | **59.7** | 62.7 | **66.2** | 75.1 | **81.7** | 73.7 | **78.4** | 68.3 | **86.1** |
| 500 | 58.6 | **58.8** | 60.0 | **60.8** | 68.9 | 66.8 | 75.7 | **81.8** | 73.0 | **80.5** | 67.0 | **86.9** |
| 1000 | 58.7 | **58.9** | 61.1 | **63.0** | 70.8 | 66.8 | 75.0 | **81.5** | 72.5 | **86.2** | 54.7 | **86.9** |
| 2000 | 58.7 | **58.9** | 63.5 | **66.7** | 69.9 | **72.0** | 75.2 | **85.7** | 69.8 | **88.0** | 44.5 | **86.6** |
| 4000 | 58.8 | **59.0** | 65.3 | **69.5** | 73.4 | **77.0** | 75.4 | **88.9** | 72.1 | **88.9** | 41.8 | **88.4** |

Table 4: We summarize the highest achieved average $\text{ESS}_p$ over the entire optimization which corresponds to the best attainable performance in terms of $\text{ESS}_p$ possible for that model capacity. The flow consists of 6 coupling layers, each with 4 linear layers, each of which has an increasing width (number of neurons).

| Layer Width | 10 | | 50 | | 100 | | 250 | | 500 | | 1000 | |
|---|---|---|---|---|---|---|---|---|---|---|---|---|
| Batch Size | Std | Path | Std | Path | Std | Path | Std | Path | Std | Path | Std | Path |
| 10 | 56.2 | **57.4** | 59.0 | **59.2** | 59.0 | **59.1** | 59.1 | **59.2** | 59.5 | **64.7** | 66.5 | **76.8** |
| 50 | 58.5 | **58.8** | 59.0 | **59.2** | 59.0 | **59.1** | 61.3 | **68.1** | 73.1 | **78.9** | 74.7 | **84.7** |
| 100 | 58.7 | **58.9** | 59.0 | **59.2** | 59.0 | **59.1** | 68.1 | **76.6** | 74.0 | **83.4** | 72.4 | **86.6** |
| 200 | 58.9 | **59.0** | 59.0 | **59.2** | 59.1 | **59.7** | 73.7 | **81.1** | 75.9 | **86.0** | 69.8 | **88.4** |
| 500 | 58.9 | **59.1** | 59.0 | **59.2** | 59.9 | **65.5** | 77.8 | **84.7** | 74.7 | **88.3** | 68.3 | **90.5** |
| 1000 | 59.0 | **59.1** | 59.1 | **59.3** | 66.8 | **73.2** | 79.1 | **85.2** | 75.7 | **90.3** | 60.4 | **88.3** |
| 2000 | 59.0 | **59.1** | 59.2 | **60.3** | 71.3 | **73.4** | 78.4 | **86.9** | 71.8 | **90.0** | 52.9 | **87.6** |
| 4000 | 59.0 | **59.1** | 59.6 | **62.9** | 73.4 | **78.0** | 77.7 | **89.0** | 68.8 | **89.9** | 50.7 | **88.0** |

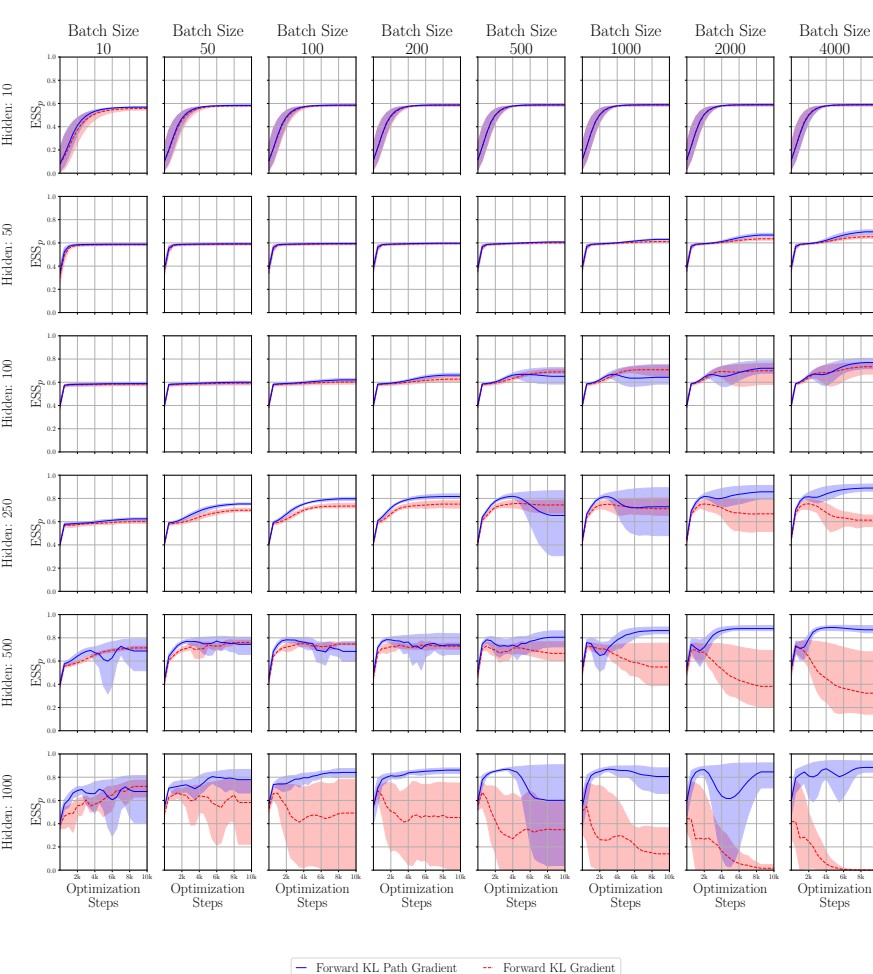

Figure 5: The $\mathrm{ESS}_p$ of a RealNVP flow with two linear layers in each of its six couplings blocks trained with Forward KL Gradients shown in red and Forward KL Path Gradients shown in blue. For higher model capacity and larger batch sizes, the Forward KL Path Gradients achieve higher absolute $\mathrm{ESS}_p$ while the Forward KL Gradients collapse with increasing model capacity. The rows increase width of the linear layers (hidden neurons) and the columns increase the batch size used during optimization.

$\phi^4$ **Field Theory.** For our flow architecture we use a slightly modified NICE (Dinh et al., 2014) architecture, called Z2Nice (Nicoli et al., 2020), which is equivariant with respect to the $\mathbb{Z}_2$ symmetry of the $\phi^4$ action in (28). We use a lattice of extent $16 \times 8$, a learning rate of 0.0005, batch size $8,000$, AltFC coupling, 8 coupling blocks with 4 hidden layers each. A learning rate decay with patience of $3,000$ epochs is applied. We used global scaling, Tanh activation and hidden width $1,000$. As base-density we chose a Normal distribution $q_0 = \mathcal{N}(0, 1)$. Gradient clipping with norm=1 is applied. Just like in Nicoli et al. (2023), training is done on 50 million samples. Optimization is performed for 48h on a single A100 each, which leeds to up to 1.5 million steps for the standard gradient estimators and 1.1 million epochs with the fast path algorithm.

$U(1)$ **Gauge Theory.** The $U(1)$ experiments are based on the experimental design in Kanwar et al. (2020). We consider a lattice with $16^2$ sites with a batch size of $12,288$, learning rate 0.0001, 24 coupling blocks with a NCP (Rezende et al., 2020) with 6 mixtures, hidden size $8 \times 8$ and kernel size 3 and a uniform base-density $q_0 = \mathcal{U}(0, 2\pi)$. We train our models on an A100 for one

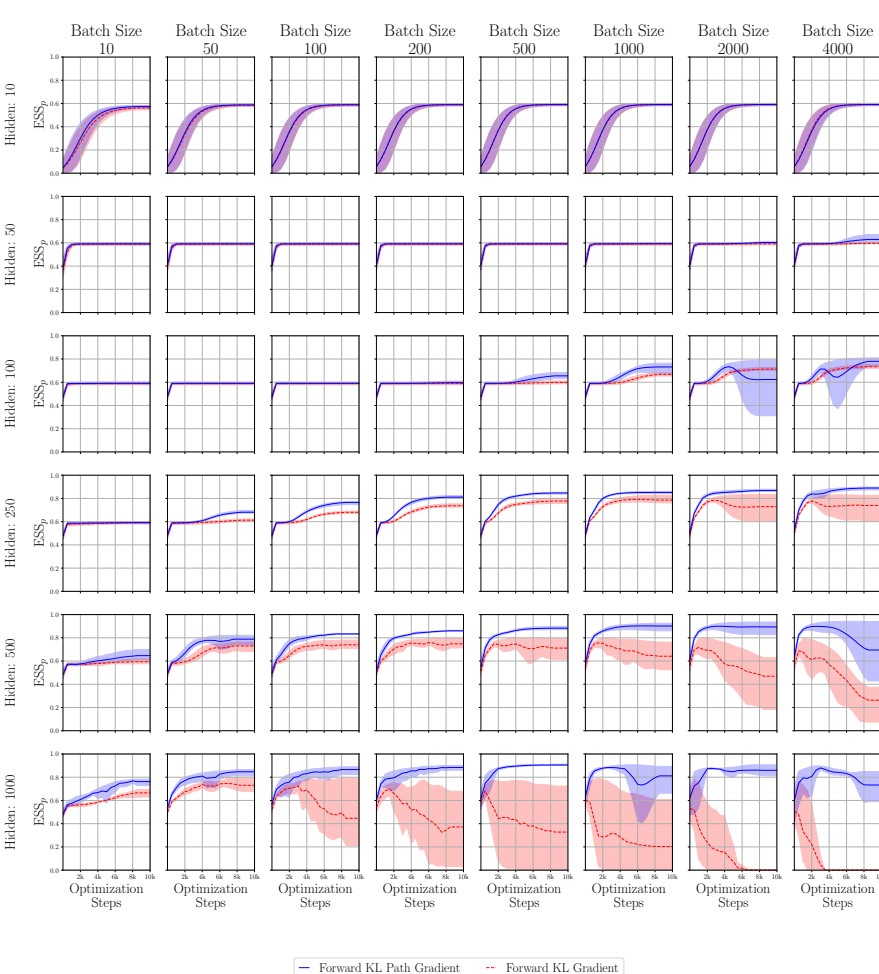

Figure 6: For many combinations of increasing parameterization and increasing batch size, Forward KL Path Gradients are able to maintain their high $\text{ESS}_p$, while performance of Forward KL Gradients increasingly deteriorates. The rows increase width of the linear layers (hidden neurons) and the columns increase the batch size used during optimization.

week, the batch size was chosen, so as to maximize GPU-RAM for a single GPU. Because training from random initialization led to very high variance in performance, we pre-train the model for $\beta = 3.0$ for 200,000 epochs. The shown benchmarks show the training after initializing from the pretrained model for target beta $\beta = 3.0$. For the standard reverse KL gradient estimator (using reparametrization trick) this yields 700,000 and for the fast path gradients 300,000 epochs.

The results can be seen in Figure 2; they are averaged over a running average of window size 3 and 4 repetitions, the mean and standard error are shown. The ESS is estimated on $10 \times$ *batch size* samples.

**Runtimes.** In order to give a fair comparison for the runtimes, we use the hyperparameters of the experiments in this work as a testing ground. Namely for the affine coupling flow, we use the $\phi^4$ experiments and for the implicitly invertible flows, we use the $U(1)$ experiments. Walltime runtimes are measured on an A100 GPU with 1,000 repetitions.

For the affine flows, we use the setup from the $\phi^4$ experiments and look at the existing Algorithm 2 for computing path gradient, the proposed fast path Algorithm 1, as well as the equally new Algo-

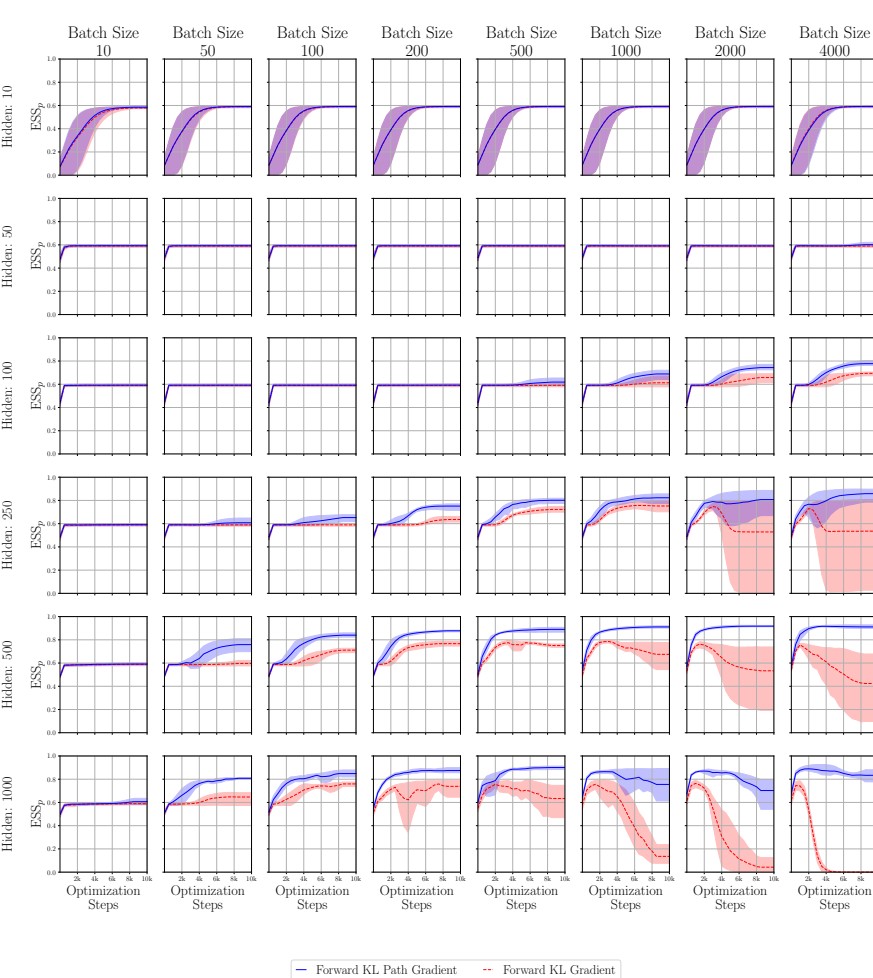

Figure 7: The higher model capacity compared to models with fewer linear layers per coupling block increases the performance of Forward KL Path Gradients in terms of $\text{ESS}_p$ while Forward KL Gradients deteriorate, resulting in a final $\text{ESS}_p = 0.0\%$ for the model with the largest capacity trained with the largest batch size. The rows increase width of the linear layers (hidden neurons) and the columns increase the batch size used during optimization.

rithm 3, which uses the insights of Section 4 to speed up Algorithm 2. Here, Algorithm 1 for the fast path gradient uses the recursive equation (49b). The results can be seen in the upper rows of Table 6.

For the implicitly invertible flows, we use the setup of the $U(1)$ experiments. Since the flows are not easily invertible, a significant percentage of the time is spent on the root finding algorithm, which is implemented as the bisection method. The root finding algorithm employs an absolute error tolerance which determines when the recursive search stops. Each iteration of the bisection method requires one evaluation of the function, in our case a normalizing flow.

Using backpropagation through the bisection is not only error-prone, but also costly in compute and memory. Recently, Köhler et al. (2021) proposed circumventing the backpropagation through the root finding via the implicit function theorem. Both of these methods can be combined with the existing path gradient Algorithm 2. Due to the increase of computational cost and numerical error of the root finding algorithm, these methods are outperformed in runtime and precision by our proposed Algorithm 1. As the tolerance in root-finding, Köhler et al. (2021) chose a value of $1e^{-6}$

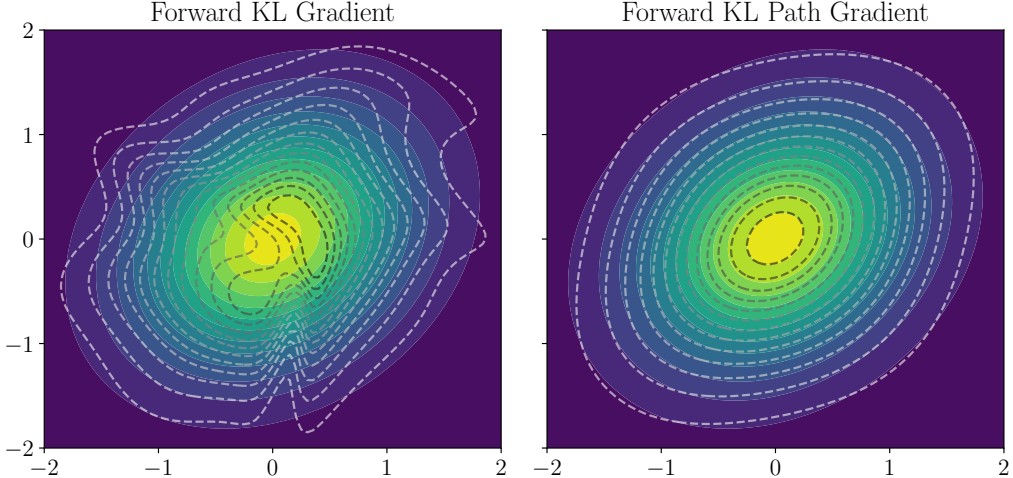

Figure 8: We visualize the contours of the learned probability distribution $q_\theta$ over the true contour plot of $p$ which is a two dimensional multivariate Gaussian distribution centered at $0$ with $0.5$ on the diagonal entries of the covariance matrix and $0.25$ on the off-diagonal entries. The RealNVP is trained with $750$ samples from $p$ with the forward KL divergence. While the forward KL Path Gradient is able to recover the true distribution well, the standard gradient exhibits numerous irregularities for the same training samples.

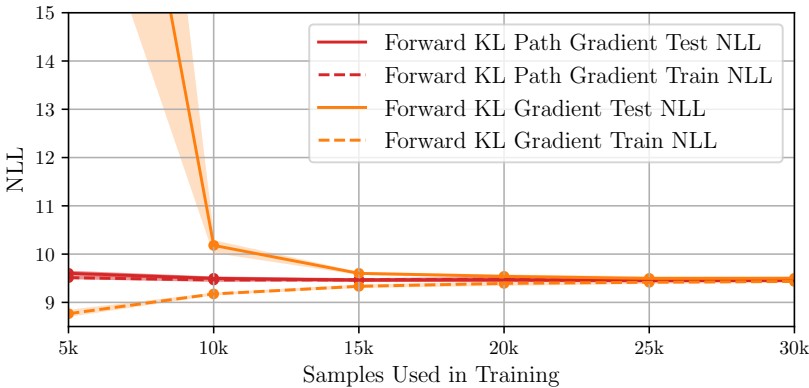

Figure 9: We plot the train and test negative Loglikelihood (NLL) intervals denote the minimum and maximum performance over 5 runs. We see that the path gradient training consistently outperforms standard training in terms of NLL. The Forward KL Path Gradients is less prone to overfitting and is able to maintain a steady test NLL. Already with comparitively few data samples, path gradients converge to the test set NLL. In data regimes with little data, forward KL gradients are prone to overfitting on the training data compared to forward KL path gradients.

Table 5: We summarize the highest achieved average $\mathrm{ESS}_p$ over the entire optimization which corresponds to the best attainable performance in terms of $\mathrm{ESS}_p$ possible for the corresponding model capacity. The flow consists of 6 coupling layers, each with 6 linear layers each of which has an increasing width (number of neurons).

| Layer Width | 10 | | 50 | | 100 | | 250 | | 500 | | 1000 | |
|---|---|---|---|---|---|---|---|---|---|---|---|---|
| Batch Size | Std | Path | Std | Path | Std | Path | Std | Path | Std | Path | Std | Path |
| 10 | 57.9 | **58.5** | 59.1 | **59.3** | 59.1 | **59.2** | 59.0 | **59.1** | 59.0 | **59.1** | 59.0 | **60.6** |
| 50 | 58.9 | **59.1** | 59.1 | **59.3** | 59.1 | **59.2** | 59.0 | **60.8** | 59.7 | **75.9** | 64.8 | **80.8** |
| 100 | 59.0 | **59.1** | 59.1 | **59.3** | 59.1 | **59.2** | 59.0 | **65.2** | 71.1 | **84.0** | 75.9 | **84.8** |
| 200 | 59.1 | **59.2** | 59.1 | **59.3** | 59.1 | **59.3** | 63.6 | **75.1** | 76.7 | **87.8** | 75.9 | **87.5** |
| 500 | 59.1 | **59.2** | 59.1 | **59.3** | 59.1 | **61.9** | 72.2 | **80.1** | 77.6 | **89.0** | 75.6 | **90.1** |
| 1000 | 59.1 | **59.2** | 59.1 | **59.2** | 61.3 | **68.9** | 75.7 | **82.3** | 78.6 | **91.1** | 75.6 | **86.5** |
| 2000 | 59.1 | **59.2** | 59.1 | **59.2** | 65.8 | **74.4** | 74.7 | **80.7** | 76.4 | **91.8** | 76.4 | **87.0** |
| 4000 | 59.1 | **59.2** | 59.1 | **60.2** | 69.4 | **77.8** | 73.1 | **85.8** | 75.7 | **91.6** | 74.6 | **88.9** |

Table 6: Factor of runtime increase in comparison to standard gradient estimator, i.e., $^{\text{runtime path gradient}}/_{\text{runtime standard gradient}}$ on A100-80GB. (For error sterr and for runtime std is shown.)

| | Algorithm | | Error | Runtime increase (batch size) | | |
|---|---|---|---|---|---|---|
| | | | $\times$ 1e-7 | 64 | 1024 | 8,192 |
| Explicitly | Alg. 1 (ours) | | - | **1.6 ± 0.1** | **1.4 ± 0.1** | **1.4 ± 0.0** |
| | Alg. 2 (Vaitl et al., 2022a) | | - | 2.1 ± 0.1 | 2.2 ± 0.1 | 2.1 ± 0.0 |
| | Alg. 3 (ours) | | - | 1.8 ± 0.0 | 1.9 ± 0.1 | 1.8 ± 0.0 |
| Implicitly | Alg. 1 (ours) | | **2.0 ± 0.4** | **2.2 ± 0.0** | **2.0 ± 0.1** | **2.3 ± 0.0** |
| | Black-box root finding | abs tol | | | | |
| | Alg. 2 + Autodiff | 2e-6 | >100,000 | 19.2 ± 0.3 | 11.8 ± 0.1 | Out of Mem |
| | Alg. 2 + Köhler et al. (2021) | 2e-6 | 29.5 ± 13.5 | 4.8 ± 0.0 | 3.4 ± 0.0 | 3.4 ± 0.0 |
| | Alg. 2 + Köhler et al. (2021) | 1e-6 | 4.1 ± 1.1 | 17.5 ± 0.2 | 11.0 ± 0.1 | 8.2 ± 0.0 |

for testing the runtime and during training on other benchmarks, they chose a tolerance of $2e^{-6}$. Results for both of these tolerances are shown in the lower rows of Table 6. Numerical errors are computed on 50 batches, each with 64 samples.

We show the behavior of the baseline Algorithm 2 with bisection and gradient computation as proposed by Köhler et al. (2021) in Figure 10. We can see that our proposed method outperforms the baseline irrespective of the chosen hyperparameters. For single-precision floating point, the numerical error can only be reduced so far, after double-precision has to be employed which leads to a drastic increase in runtime and memory.

# F    CONTRIBUTIONS

| | LV | LW |
|---|---|---|
| Conceptualization | × | |
| Methodology | × | |
| Formal Analysis | × | |
| Software | × | × |
| Investigation | × | × |
| Visualization | × | × |
| Writing - Original Draft | × | × |
| Writing - Review & Editing | × | |

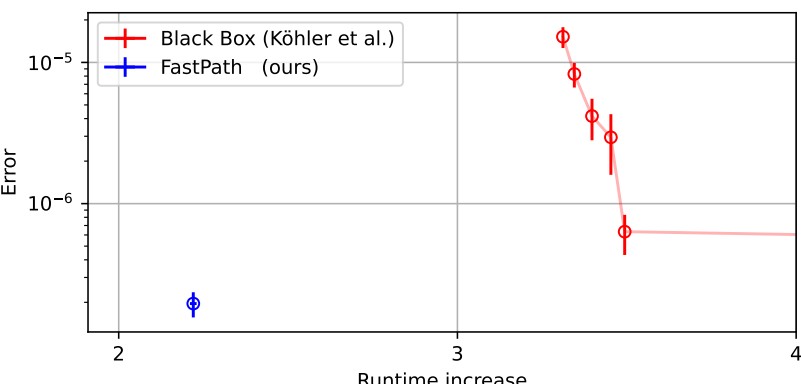

Figure 10: Runtime vs Precision trade-off in Köhler et al. (2021) $U(1)$ experiments with lattice $16^2$

