# OpenReview forum: "Fast and unified path gradient estimators for normalizing flows"
_ICLR.cc/2024/Conference — ICLR 2024 poster_

### Official Review · Reviewer_ZWTi · 2023-10-27

**Soundness:** 2 fair
**Presentation:** 2 fair
**Contribution:** 2 fair
**Rating:** 8
**Confidence:** 4

**Summary:**

The paper concerns path gradient estimators for normalizing flows, a reduced variance estimator for the KL divergence gradient in normalizing flows. They come at the cost of additional forward and backward passes through the normalizing flow at hand. The present paper reduces this computational overhead to compute path gradient of reverse KL, while being analytically equal to previous work. It then provides a new path gradient estimator for the forward KL. Experiments demonstrate that the resulting path gradient estimators work both in the forward and reverse KL setting on physical sciences data sets (where the unnormalized $p(x)$ is known).

**Strengths:**

*Originality*

- The iterative procedure for computing the path gradient has no memory overhead over non-path gradients and is potentially faster (see Weakness 3).
- Path gradients are applied to the forward KL with reduced variance by applying the same algorithm to .
- The approach has the potential to be generically applied to abitrary coupling blocks, if clarified.


*Quality*

The theoretical results might be correct, but I cannot judge at this point (see below). I have some doubts on the baseline experiments (see below).


*Clarity*

The motivation and main chain of reasoning are clear, but several parts of the manuscript lack clarity and detailed explanations (see below).


*Significance*

Making use of path gradients in order to regularize for the known unnormalized density of training data has the potential to greatly reduce compute over classical methods, so this chain of work is relevant to the machine learning + natural sciences community. Allowing the forward KL to make use of the unnormalized density is attractive, as the forward KL may have better properties than reverse KL (mode covering instead of mode seeking).

**Weaknesses:**

Generally, the presentation interpretation of the results can be greatly improved. I also have concerns on some of the results.

In detail:

1. The notation of Proposition 3.2 and its proof in the appendix are sloppy and I cannot determine the correctness: what is the inverse of the rectangular matrix $\frac{\partial f_\theta(x_l^t, x_l^c)}{\partial x_l^t}$? Is it a pseudo-inverse, or is it a part of the network Jacobian? I suggest to greatly rewrite this proposition as a Theorem that outlines the general idea of the recursion (that the path gradient can be constructed iteratively by vector-Jacobian products with the inverse of each block, if I am right). Then proceed to derive concrete realizations for coupling blocks and affine couplings in particular if they allow for unique results.
2. What is the cost of computing Proposition 3.2? As I mentioned in the first point, by rewriting the recursion more generally, this could easily be showcased.
3. What is the intuition behind Proposition 4.1? What is the regularization obtained from including the unnormalized density (probably something like the corrected relative weight of each sample according to the ground truth density)?
What derivative vanishes in expectation? How large is the variance of the removed gradient term? Is your result an improvement in this metric? What is the regularizing effect? Vaitl et al. 2022b have useful visualizations and explanations in this regard.
4. The baseline Algorithm 2 should not be used to measure baseline times. The second forward pass through the network is unneccessary, as one can simply store the result from the first forward pass, once with stop_gradient and once without. Please report Table 2 again with this change.
5. I have strong doubts on the validity of the experiment on the multimodal gaussian model. It is hard to believe that a standard RealNVP network cannot be trained effectively on this data, with an ESS_p of 0.0(!). I see several warning signs that a bad performing network has been selected in order to have a badly performing baseline:
	- the network is huge, with a number of parameters bounded from below by six coupling blocks $\times$ five hidden subnetworks $\times$ (1000 $\times$ 1000 entries in each weigh matrix) amounting to more than 30 million parameters;
	- the batch size of 4,000 given 10,000 samples makes the network see almost the entire data set in every update.
  This indicates that the training is set up in a way that training from samples only must fail. Given that training yields useful models in only five minutes, it is reasonable to expect hyperparameter tuning of the baseline model from the authors.
6. In this light, how much parameter tuning was involved in the other experiments $\phi^4$ and $U(1)$? Please compare your numbers to the state of the art results on these benchmarks.


Given that the theoretical results need improved presentation and explanation, and given the doubts on the numerical experiments, the manuscript does not reach the quality ICLR in the current form. Many of the proposed changes can be achieved with additional explanations and better notation. I am looking forward to the author's rebuttal, happy to be corrected on my understanding.



## Minor comments:

- Eq. (13) is missing a logarithm.
- The caption for Figure 1 is on page 21 in the appendix, took me some time.
- The statement that building up the computation graph takes measurable time is false, as this simply means storing already  computed activations in a dictionary (right before section 3.1).
- Eq. (25) is missing that $p_{\theta, 0}$ can be computed from the unnormalized density.
- If a reader is not familiar with the terms forward and reverse KL, it is hard to understand the introduction. Point the reader to Section 2 or drop it here, leaving space for more explanations on theoretical results.

**Questions:**

see Weaknesses.

---

> ### Author Response · Authors · 2023-11-21
>
> ### Strengths
>
> ***Originality***
>
> - **The iterative procedure for computing the path gradient has no memory overhead over non-path gradients and is potentially faster (see Weakness 3).**
> - **Path gradients are applied to the forward KL with reduced variance by applying the same algorithm to .**
> - **The approach has the potential to be generically applied to abitrary coupling blocks, if clarified.**
>
> ***Quality***
>
> **The theoretical results might be correct, but I cannot judge at this point (see below). I have some doubts on the baseline experiments (see below).**
>
> ***Clarity***
>
> **The motivation and main chain of reasoning are clear, but several parts of the manuscript lack clarity and detailed explanations (see below).**
>
> ***Significance***
>
> **Making use of path gradients in order to regularize for the known unnormalized density of training data has the potential to greatly reduce compute over classical methods, so this chain of work is relevant to the machine learning + natural sciences community. Allowing the forward KL to make use of the unnormalized density is attractive, as the forward KL may have better properties than reverse KL (mode covering instead of mode seeking).**
>
> We thank the reviewer for the positive feedback. We agree that our method is generically applicable to coupling-based flows and is of relevance to natural sciences applications.
>
> ### Weaknesses
>
>  **1. *The notation of Proposition 3.2 and its proof in the appendix are sloppy and I cannot determine the correctness: what is the inverse of the rectangular matrix
>  $\partial f_{\theta}(x_{l}^{trans}, x_{l}^{cond})/\partial x_{l}^{trans}$
>  ? Is it a pseudo-inverse, or is it a part of the network Jacobian? I suggest to greatly rewrite this proposition as a Theorem that outlines the general idea of the recursion (that the path gradient can be constructed iteratively by vector-Jacobian products with the inverse of each block, if I am right). Then proceed to derive concrete realizations for coupling blocks and affine couplings in particular if they allow for unique results.***
>
> The Jacobian matrix  $\frac{\partial f_\theta(x_l^{trans}, x_l^{cond})}{\partial x_l^{trans}}$ is square and invertible and there is thus no subtlety in defining its inverse. In more detail, we define a coupling block in Eq 6 as
>
>  $$x_{l+1}^{trans} = f_{\theta} (x_{l}^{trans}, x_{l}^{cond})$$
>
>  $$x_{l+1}^{cond} = x_{l}^{cond}$$
>
> where $f_\theta(\bullet \,, x_l^{cond})$ is an invertible function for any choice of $x_l^{cond}$.
> By bijectivity, the Jacobian matrix from above is not only square but also invertible.
> We have added a remark after Proposition 3.3 to emphasize this and we have rewritten the proof, so that this becomes more clear.
>
> We have revised the manuscript implementing your suggestions.
> Specifically, we have added Proposition 3.2, where we first state the recursion for a general flow, i.e., not necessarily a coupling flow:
>
> $\frac{\partial \log q_{\theta,  l+1}(x_{l+1})}{\partial  x_{l+1}} = \frac{\partial \log q_{\theta, l}(x_l)}{\partial  x_{l}} \left( \frac{\partial T_{l,  \theta_l}(x_l)}{\partial x_l} \right)^{-1} + \frac{\partial \log |\det \frac{\partial T_{l, \theta_l}(x_l)}{\partial x_l}|}{\partial x_l}\left( \frac{\partial T_{l, \theta_l}(x_l)}{\partial x_l} \right)^{-1}$
>
> As we also remark in the modified manuscript, the evaluation of this expression however involves inversion of the Jacobian and is therefore prohibitively expensive for generic flow architectures (see answer immediately below for more details).

---

> > ### Author Response · Authors · 2023-11-21
> >
> > ***2. What is the cost of computing Proposition 3.2? As I mentioned in the first point, by rewriting the recursion more generally, this could easily be showcased.***
> >
> > First, note that in the revised manuscript Proposition 3.2 has become Proposition 3.3. We will refer to proposition numbers in the revised manuscript.
> >
> > For general flow architectures the recursion formula is given by the new Proposition 3.2. For generic architectures, where the inverse of the transformation $T_{l, \theta_l}$ is not easy to compute, this recursion however is too expensive. Specifically it involves building and inverting of a $d\times d$ Jacobian, which naively scales cubically in the number of dimensions $d$ (although optimized algorithms with slightly better scaling exist).  Even for autoregressive flows, which have triangular Jacobians, there is still a quadratic scaling. This compares unfavorable with the linear scaling of the standard total gradient estimator. For this reason then we derived a specialized recursion for coupling flows (see Proposition 3.3). This has the same linear complexity as the standard gradient estimators.
> > Furthermore in Corollary 3.4 we explicate for affine coupling blocks, which has the same linear complexity, but is slightly faster, since it reuses terms that have been used in the standard forward pass and uses less terms than the recursion in Proposition 3.3.
> >
> > In our experiments, we  used the recursion from Proposition 3.3 for implicitly invertible flows, while Corollary 3.4 was used for explicitly invertible flows. The walltime cost of computation is summarized in Table 2 (for Proposition 3.3 see third row 'Implicitly Invertible' and for Corollary 3.4 see first row, 'Explictly Invertible').
> > Computing Proposition 3.3 has roughly twice the computational cost of the standard total gradient estimator, but compares favorable to the previous state of the art in the literature, which under the same constraints (numerical error and memory footprint) yields a time factor of at least 8.
> > Even if we relax the numerical constraints in favor of the baseline, our fast estimators still lead to significantly faster computations, i.e. at least a 33\% speed up with an order of magnitude higher precision.

---

> > > ### Author Response · Authors · 2023-11-21
> > >
> > > **3. What is the intuition behind Proposition 4.1? What is the regularization obtained from including the unnormalized density (probably something like the corrected relative weight of each sample according to the ground truth density)? What derivative vanishes in expectation? How large is the variance of the removed gradient term? Is your result an improvement in this metric? What is the regularizing effect? Vaitl et al. 2022b have useful visualizations and explanations in this regard.**
> > >
> > > We substantially extended Appendix B.3 to address your questions in detail. Briefly summarized:
> > >
> > > - Intuition: It is well known that the forward KL in target space becomes the reverse KL in base space (see Section 2.3.3 in [1]). This duality allows us to immediately apply all the results derived for the reverse KL case to the forward case, if we express all densities in the base space variables. To the best of our knowledge, we are the first to observe and harness this simple, yet powerful trick.
> > > - Regularization: We discuss this in detail in the new subsection B.3.2. In essence, the path gradient estimator involves a term containing the derivative of the log of the target density, $\nabla_{x_0} \log p(T_\theta(x_0)) \frac{\partial T_\theta^{-1}(x)}{\partial \theta}$, see Eq. 57. This term is absent in the standard total gradient, see Eq. 56. Note that in the case of Boltzmann generators, this derivative of the log target density is known in closed form by assumption. Thus, the path gradient estimator is able to harness information of the gradient of the energy at the given sample, which can be thought of as a (gradient) regularizer.
> > > - Term vanishing in expectation: The expected score of the pullback target density $E_{p_{0, \theta}}[\frac{\partial \log p_{0, \theta}(x_0)}{\partial \theta}]$ vanishes in expectation. This term has non-vanishing variance in the standard gradient estimator, even if the pullback $p_{0, \theta}(x_0)$ perfectly approximates the base density $q_0$
> > > (see the revised Appendix B.3).
> > > - Variance of the removed term: The removed variance corresponds to the Fisher Information Matrix (divided by the batch size) of the pullback density $p_{0, \theta}$ (see Appendix B3.1).
> > > - Improvement in metric: We observe exactly the same beneficial sticking-the-landing behavior of our forward estimators as observed in the reverse case, i.e. the variance of the estimator vanishes if the model perfectly approximates the target and this metric is overall reduced particularly at the end of training (see Figure 3 in the revised manuscript).
> > > - Useful visualization: We added the exact same plot as in Vaitl et al. 2022b (see Figure 3 and analogous explanations in the revised appendix B3).
> > >
> > > [1] Papamakarios, George, et al. "Normalizing flows for probabilistic modeling and inference." The Journal of Machine Learning Research 22.1 (2021): 2617-2680. https://arxiv.org/abs/1912.02762
> > >
> > > **4. The baseline Algorithm 2 should not be used to measure baseline times. The second forward pass through the network is unneccessary, as one can simply store the result from the first forward pass, once with stop_gradient and once without. Please report Table 2 again with this change.**
> > >
> > > Your concern is valid but addressed in our manuscript.
> > > In more detail, your proposed algorithm is applicable but has substantially higher memory footprint because it cannot discard the computational graph for $x'$ of equation 12 in memory.
> > > This is often prohibitive, for example in the context of lattice field theory, as large batch sizes are essential for successfull training. As a result, the path gradient literature, such as Vaitl et al., focuses on estimators with a comparable memory footprint.
> > > We added Figure 11 to the appendix, which experimentally measures the substantial increase in memory footprint of your proposal.
> > >
> > > However, along similar lines, we propose a different but novel baseline Alg. 3 in the appendix of our paper which not only has the same runtime as your proposal but also has comparable memory cost as our fast gradient estimators.
> > > We emphasize that this proposal is an original contribution of our manuscript which outperforms the previous state-of-the-art, i.e., Alg. 2 as proposed by Vaitl et al. Detailed runtime comparisions to this novel baseline, Alg 3, as well as to the previous state-of-the-art, Alg 2, can be found in Table 3 and clearly establishes that we outperform both baselines across the board by at least a speedup of 22%.
> > > We refrained from discussing this in the main text as introducing a novel baseline only to then show that it is beaten by another novel estimator would lead to a convoluted presentation.

---

> > > > ### Author Response · Authors · 2023-11-21
> > > >
> > > > **5. I have strong doubts on the validity of the experiment on the multimodal gaussian model. It is hard to believe that a standard RealNVP network cannot be trained effectively on this data, with an ESS_p of 0.0(!). I see several warning signs that a bad performing network has been selected in order to have a badly performing baseline:**
> > > >
> > > > - **the network is huge, […]**
> > > > - **the batch size of 4,000 given 10,000 samples makes the network see almost the entire data set in every update. […]**
> > > >
> > > > We politely disagree with the statement that our experiments supposedly show that RealNVP cannot learn a MGM. Rather, our experiments merely establish that path gradients facilitate *more sample efficient training* and help avoid overfitting.
> > > > This can be seen from the rhs of Fig 1, namely that standard maximum likelihood training can easily fit the MGM if enough samples are provided. For smaller number of samples, maximum likelihood training leads to overfitting. A non-vanishing ESS_p could be achieved by early stopping. Early stopping is however challenging in the context of Boltzmann generators, as their training often relies on a limited number of biased samples which are only used for pre-training. In the revised manuscript, we nevertheless report results with early stopping to adopt the most charitable setting for the baseline. We stress that sample efficiency is a crucial requirement in learning unnormalized distributions for which training set generation involves costly MD or MCMC simulations.
> > > >
> > > > To avoid any impression of hyperparameter tuning and to provide a more detailed analysis, we include an extensive analysis of results for various batch and flow sizes in the revised Appendix E. In more detail, Figures 5, 6 and 7 show the corresponding Forward ESS during the course of both path and non-path forward training for varying numbers of
> > > > - linear layers per coupling block,
> > > > - number of hidden neurons per linear layer,
> > > > - batch sizes.
> > > >
> > > > While standard Forward KL gradients can *relatively* surpass Forward KL Path Gradients for smallest models trained with the two smallest batch sizes by a small margin, a larger model trained with path gradients doubles the mean of the $ESS_p$ in a direct comparison and improves upon the best forward KL gradients by almost 10 percentage points in absolute terms.
> > > >
> > > > Tables 3, 4 and 5 summarize the best possible Forward ESS over the course of training (corresponding to the results obtained by early stopping) for direct comparison.
> > > >
> > > > We have rephrased the relevant experimental section to make this point clearer.

---

> > > > > ### Author Response · Authors · 2023-11-21
> > > > >
> > > > > **6. In this light, how much parameter tuning was involved in the other experiments $\phi^4$ and U(1) ? Please compare your numbers to the state of the art results on these benchmarks.**
> > > > >
> > > > > For our $\phi^4$ experiments, we use the exact same hyperparameter choices as in the recent publication [1] as well as the same codebase.
> > > > > To the best of our knowledge, there are currently only two publications considering maximum likelihood training for $\phi^4$ theory: [1] and [2].
> > > > > We chose [1] as it is more recent.
> > > > >
> > > > > For the target density, we choose the same bare quartic coupling $\lambda = 0.022$ as in [1] and hopping parameter $\kappa = 0.25$.
> > > > > The authors of [1] considered various hopping parameters and we restrict to this particular choice as it is the most physically relevant one: $\phi^4$ theory has a second order phase transition at this critical point, see Figure 1 in [2].
> > > > > It is well-known that second order phase transitions lead to divergent correlation length and thus to long-range correlation which are challenging to learn.
> > > > > Unfortunately, [1] and [2] do not report any effective sampling sizes to compare to. However, we confirmed in private communication with the authors of [1] that our effective sampling sizes of 85.1% for the baseline are compatible with their results. We also experimentally calculated the wbar estimator which is 1.0001+-8e-5 and thus compatible with Figure 21 in [1] for the considered target parameters.
> > > > >
> > > > > For U(1), we base our experiments on [4] using their code base (which was released in this separate publication [5]).
> > > > > To the best of our knowledge, this is the only publicly available code for a normalizing flow with U(1) gauge symmetry.
> > > > > The chosen bare coupling parameter is $\beta=3$ as in this publication (in addition, the authors also fine-tuned from this target value to further bare coupling values).
> > > > > We followed the same hyperparameter choices as in the publication.
> > > > > Unfortunately, important hyperparameter choices such as details on the architecture and training setup were not reported in the publication and we therefore had to rely on the PhD thesis [5] of one of the authors to extract more details.
> > > > > We choose exactly the same architecture and hyperparameteres as listed in the thesis except for the batch size and learning rate.
> > > > > This is because the thesis is not clear on the precise batch size stating that “batches of size ranging from 16,384 to 131,072” were used, see Section 4.6.3 on page 118.
> > > > > We used a batch size of 12,288 since it was the biggest one that we could reach without gradient accumulation.
> > > > > We also chose a learning rate of 1e-4 as opposed to 1e-3 reported in the thesis since training proved unstable with a higher learning-rate (most likely due to the difference in batch size).
> > > > > Furthermore, [4] does not seem report any effective sampling sizes to compare to but our results are consistent with the integrated autocorrelation time shown in Figure 4 of [4].
> > > > > Note that in neural MCMC, the integrated autocorrelation time is closely related to the effective sampling size, see [https://arxiv.org/abs/2111.10189](https://arxiv.org/abs/2111.10189) for a rigorous discussion.
> > > > >
> > > > > [1] Detecting and Mitigating Mode-Collapse for Flow-based Sampling of Lattice Field Theories, Nicoli et al. 2023, [https://arxiv.org/abs/2302.14082](https://arxiv.org/abs/2302.14082)
> > > > >
> > > > > [2] Flow-based sampling for multimodal distributions in lattice field theory, Hacket et al, 2021, [https://arxiv.org/abs/2107.00734](https://arxiv.org/abs/2107.00734)
> > > > >
> > > > > [3] Estimation of Thermodynamic Observables in Lattice Field Theories with Deep Generative Models, Nicoli et al, [https://journals.aps.org/prl/abstract/10.1103/PhysRevLett.126.032001](https://journals.aps.org/prl/abstract/10.1103/PhysRevLett.126.032001)
> > > > >
> > > > > [4] Equivariant flow-based sampling for lattice gauge theory, Kanwar et al,  Physics Review Letters 2020, [https://arxiv.org/abs/2207.08945](https://arxiv.org/abs/2003.06413)
> > > > >
> > > > > [5] Introduction to Normalizing Flows for Lattice Field Theory, Albergo et al, [https://arxiv.org/abs/2101.08176](https://arxiv.org/abs/2101.08176)
> > > > >
> > > > > [6] Machine Learning and Variational Algorithms for Lattice Field Theory, Gurtej Kanwar [https://arxiv.org/abs/2106.01975](https://arxiv.org/abs/2106.01975)
> > > > >
> > > > > [7]Analysis of autocorrelation times in Neural Markov Chain Monte Carlo simulation, Bialas et al, 2023, Physical Reviews E, [https://arxiv.org/abs/2111.10189](https://arxiv.org/abs/2111.10189)

---

> > > > > > ### Author Response · Authors · 2023-11-21
> > > > > >
> > > > > > ### Minor comments
> > > > > >
> > > > > > **Eq. (13) is missing a logarithm.**
> > > > > >
> > > > > > We fixed the typo.
> > > > > >
> > > > > > **The caption for Figure 1 is on page 21 in the appendix, took me some time.**
> > > > > >
> > > > > > We rephrased the caption.
> > > > > >
> > > > > > **The statement that building up the computation graph takes measurable time is false, as this simply means storing already computed activations in a dictionary (right before section 3.1).**
> > > > > >
> > > > > > Thank you for pointing this out. We have deleted the statement.
> > > > > >
> > > > > > **Eq. (25) is missing that  can be computed from the unnormalized density.**
> > > > > >
> > > > > > We moved the corresponding remark into the proposition.
> > > > > >
> > > > > > **If a reader is not familiar with the terms forward and reverse KL, it is hard to understand the introduction. Point the reader to Section 2 or drop it here, leaving space for more explanations on theoretical results.**
> > > > > >
> > > > > > We have pointed the reader to Section 2 in the introduction as suggested.
> > > > > >
> > > > > > In conclusion, we want to thank you again for your extensive review, which helped us to improve our manuscript. We would be happy if you considered reevaluating your score. Please let us know if you have any further concerns or questions.

---

> > > > > > > ### Comment · Reviewer_ZWTi · 2023-11-22
> > > > > > >
> > > > > > > I thank the authors for the significant improvements to the paper. In the following, I come back to the points that I still have questions about. Please consider all other points as adequately addressed by your answers and updated manuscript.
> > > > > > >
> > > > > > > ### 3. Forward path gradient
> > > > > > >
> > > > > > > Thank you for giving more details on the gradient estimate and deriving the sticking-the-landing property, I think this already makes a compelling argument for the use of path gradients for the forward KL. I am confused by Appendix B.3.2 as its title promises beneficial regularization, but really, the equation only motivates a conjecture. Please only promise what you can hold.
> > > > > > >
> > > > > > > ### 5. Experimental validity
> > > > > > >
> > > > > > > Thanks for trying more hyper parameters. This verifies that the original hyper parameters are particular beneficial for the path gradient setting and particularly bad for standard gradients (Figure 1 seems to show the worst setup in the ablation!).
> > > > > > >
> > > > > > > The following concerns remain for me:
> > > > > > >
> > > > > > > 1. Standard gradients are only applied to a minimal batch size of 500, but decreasing batch size seems to improve results. Can you add additional runs with batch sizes 50, 100, and 200? The current data looks promising, but I think that additional data points are missing. In particular, this could prevent overfitting.
> > > > > > > 2. Figure 1 is not updated and shows a spuriously bad baseline in the light of the new data. This may be a mistake since the authors state that they want to update the figure caption, but the caption still does not explain the figure.
> > > > > > > 3. Why is early stopping difficult? Isn't the ESS a metric that can be evaluated any time during training?
> > > > > > >
> > > > > > > ### 6. $\phi^4$ experiments
> > > > > > >
> > > > > > > I think that the choice of this experiment is not optimal given that there are other Boltzmann generator datasets with published ESSs to compare to.
> > > > > > >
> > > > > > > I again thank the authors for their thorough answers. When the above points are addressed, I will reevaluate my score.

---

> > > > > > > > ### Author Response · Authors · 2023-11-22
> > > > > > > >
> > > > > > > > Thank you for your further questions and we would like to respond as follows:
> > > > > > > >
> > > > > > > > **3. Forward path gradient**
> > > > > > > >
> > > > > > > > **Thank you for giving more details on the gradient estimate and deriving the sticking-the-landing property, I think this already makes a compelling argument for the use of path gradients for the forward KL. I am confused by Appendix B.3.2 as its title promises beneficial regularization, but really, the equation only motivates a conjecture. Please only promise what you can hold.**
> > > > > > > >
> > > > > > > > We fully agree and have changed the title to “Motivation for regularization”. We welcome other suggestions if you feel this is not a good title.
> > > > > > > >
> > > > > > > > **5. Experimental validity**
> > > > > > > >
> > > > > > > > **Thanks for trying more hyper parameters. This verifies that the original hyper parameters are particular beneficial for the path gradient setting and particularly bad for standard gradients (Figure 1 seems to show the worst setup in the ablation!).**
> > > > > > > >
> > > > > > > > We have updated Figure 1. Now the central plot shows the performance using the optimal hyperparameters for both gradient estimators. We have highlighted training until early stopping. We have also updated the rightmost plot, so that the best performance is shown, instead of the final one.
> > > > > > > >  We are open to other suggestions of how to summarize the findings of the appendix.
> > > > > > > >
> > > > > > > > **The following concerns remain for me:
> > > > > > > > Standard gradients are only applied to a minimal batch size of 500, but decreasing batch size seems to improve results. Can you add additional runs with batch sizes 50, 100, and 200? The current data looks promising, but I think that additional data points are missing. In particular, this could prevent overfitting.**
> > > > > > > >
> > > > > > > > We have added the results for the requested batch sizes in Figures 5-7 and Tables 3-5. Please let us know if you have any further requests.
> > > > > > > >
> > > > > > > > **Figure 1 is not updated and shows a spuriously bad baseline in the light of the new data. This may be a mistake since the authors state that they want to update the figure caption, but the caption still does not explain the figure.**
> > > > > > > >
> > > > > > > > Our apologies. This was indeed a mistake which we fixed now.
> > > > > > > >
> > > > > > > > **Why is early stopping difficult? Isn’t the ESS a metric that can be evaluated any time during training?**
> > > > > > > >
> > > > > > > > Thanks for the question. We need to provide some context first: the forward ESS is the relevant metric when a Boltzmann generator is used. This is because we want to reweight the samples with respect to the target distribution to ensure asymptotic unbiasedness (or more precisely statistical consistency). The variance of the importance weight estimator
> > > > > > > > $\frac{1}{N} \sum_{i=1}^N w(x_i) f(x_i) $
> > > > > > > > where $w(x)=\frac{p(x)}{q(x)}$ is the importance weight of the target $p$ with respect to the sampler $q$, and $f$ is the observable of interest, such as energy or magnetization.
> > > > > > > > The variance of the estimator is given by $\frac{Var(f)}{N \cdot ESS}$ for $N$ samples from $q$. Thus, the effective sampling size directly controls the uncertainty of the estimate.
> > > > > > > >
> > > > > > > > The effective sampling size is, however, a tricky quantity to evaluate with good precision. This is for several reasons: first, the dataset could miss modes. This is a real concern in LQFT where the modes would correspond to sectors of different topological charge and MCMC-based sampling often struggles with jumping between topological sectors. This effect is known as topological freezing. A similar situation arises in proteins where some conformations may not be covered. Then, the estimated ESS on the dataset would be vastly overestimated. A similar effect arises in the tail regions of the distribution. If it holds that $q << p$ but $p$ is still small, then the ESS would also be very low. As such, we need a dataset that is big enough to resolve the tail in sufficent resolution. Finally, forward training is often only used for pre-training purposes for further energy-based training. Energy-based training relies on self-sampling. In high-dimensions, this will not work for a randomly initialized flow as the modes of the target are increasingly concentrated. One thus uses, for example, short MCMC simulations around known modes and combines them to obtain a biased dataset for pretraining. This then leads to a “warm initalization” that allows the self-sampling to probe relevant regions of sampling space. Note that the pretraining dataset is likely biased (for example the weighting between the modes is incorrect). Therefore, the effective sampling size with the biased dataset will most likely be rather different from the effective sampling size with respect to the true target. Note that we want to be sure that we avoid overfitting in this situation as we merely want to highlight regions of sampling space for further exploration by self-sampling.
> > > > > > > > While we realize that this is a bit of a lengthy answer, the above reasons explain why ESS is a metric that has to be understood with care and can not easily be used for early stopping. Please let us know if you have any further questions regarding this point.

---

> > > > > > > > > ### Author Response · Authors · 2023-11-22
> > > > > > > > >
> > > > > > > > > **6. $\phi^4$ experiments**
> > > > > > > > >
> > > > > > > > > **I think that the choice of this experiment is not optimal given that there are other Boltzmann generator datasets with published ESSs to compare to.**
> > > > > > > > >
> > > > > > > > > We have to concede this point. At the time of writing, we were under the impression that code for their experiments would be available at this stage. The paper has unfortunately just been accepted at Physical Review D and the authors have decided to release their code only after publication. We will add the link to the camera ready version. We also encourage the reviewer to reach out the authors of [1]  to confirm that our ESS values are aligned with their results. We also note that for the very same target density ($\kappa$=0.25, $\lambda$=0.022 and 16$\times$8 lattice), effective sampling sizes were given in [2]: their result of $\sim$60\% (green line in Figure 9) compares favorable with our result of 85.6\%. Note however that they used a  coupling based Stochastic Normalizing Flow.
> > > > > > > > >
> > > > > > > > > [1] Detecting and Mitigating Mode-Collapse for Flow-based Sampling of Lattice Field Theories, Nicoli et al. 2023, https://arxiv.org/abs/2302.14082
> > > > > > > > >
> > > > > > > > > [2] Caselle, Michele, et al. "Stochastic normalizing flows as non-equilibrium transformations." Journal of High Energy Physics 2022.7 (2022): 1-31. https://link.springer.com/article/10.1007/JHEP07(2022)015

---

> > > > > > > > > > ### Comment · Reviewer_ZWTi · 2023-11-22
> > > > > > > > > >
> > > > > > > > > > Thanks to the authors for the updates, the experiments are convincing now. The explanation on the details of the effective sample size has also been very illustrative to me.
> > > > > > > > > >
> > > > > > > > > > I now think that the paper is a valuable contribution to the community and recommend acceptance.

---

> > > > > > > > > > > ### Author Response · Authors · 2023-11-22
> > > > > > > > > > >
> > > > > > > > > > > Thank you for the very useful exchange which helped us to improve our manuscript.

---

### Official Review · Reviewer_UvVP · 2023-10-31

**Soundness:** 3 good
**Presentation:** 3 good
**Contribution:** 3 good
**Rating:** 6
**Confidence:** 3

**Summary:**

The authors propose a technique for improving the efficiency of the calculation of path-gradients for both the forwards and reverse KL loss.
Typically, the path gradient is lower variance but has a significantly higher computational cost, preventing scalability to large problems. Their method avoids having to evaluate the flow in both the forwards and reverse directions by recursively calculating the gradient during the forward pass using JVPs.
The speedup is especially significant for flows that require implicit differentiation for inversion. The main contributions are (1) efficient calculation of the path gradient based losses and (2) path gradient version of the forwards KL loss.

**Strengths:**

- The method obtains significant improvement in speed in practice, especially for the case of flows that require implicit differentiation for inversion.
- The method obtains improved generalization for the forward KL training relative to
- Incoporating the energy function of the target in the forward KL training is novel. And having a loss with the “sticking the landing” property for the forward KL is useful.

**Weaknesses:**

- The speedup for explicitly invertible flows (which are more common) is relatively minor.
- The authors emphasise that an advantage of their method relative to those from Vaitl et al. for the estimation of the forward KL is that their method does not require reweighting. However, their method uses samples from the target, while the method from Vaitl et al. uses samples from the flow - hence the two methods are not directly comparable as they are for different situations. I think this is somewhat misleadingly presented in the text (it is presented as an improvement relative to the forward KL objective from Vaitl).

**Questions:**

- How come the flow trained via the standard maximum likelihood objective achieves such poor performance on the MGM problem (Table 1)?. It seems possible that poor hyper-parameters have been used as training by maximum likelihood should be able to obtain reasonable results.

- In the case of forwards KL with flows that require implicit differentiation for inversion, is it not more efficient to set the forwards direction of the flow to map from the target to the flow’s base (rather than base to target), such that implicit differentiation is required for sampling, but not density evaluation)?

---

> ### Author Response · Authors · 2023-11-21
>
> ### Strengths
>
> - **The method obtains significant improvement in speed in practice, especially for the case of flows that require implicit differentiation for inversion.**
> - **The method obtains improved generalization for the forward KL training relative to**
> - **Incoporating the energy function of the target in the forward KL training is novel. And having a loss with the “sticking the landing” property for the forward KL is useful.**
>
> We thank the reviewer for recognizing these strengths. We fully agree that regularization of maximum likelihood training with the ground-truth energy function is an important contribution of our manuscript. We also appreciate that the reviewer rightly pointed out the significance of path gradients for implicitly invertible normalizing flows in the context of Boltzmann generators.
>
> ### Weaknesses
>
> **The speedup for explicitly invertible flows (which are more common) is relatively minor.**
>
> Our estimators for explicitly invertible flows have about 60 percent the runtime of the previous state-of-the-art. Thus the speed-up is significant. We however agree that proposing path gradient estimators for implicitly invertible normalizing flows is probably the more important contribution of our manuscript since many state-of-the-art architectures in the Boltzmann generator context are of this form. Particular examples are Smooth Flows [1] for Quantum Chemistry as well as the gauge invariant coupling flows [2-4]. To the best of our knowledge, we are the first to propose fast path gradient estimators for this model class of high practical relevance. A further important contribution of our manuscript is that we provide the first path-gradient estimators for sample-based training (see discussion below).
>
> [1] Smooth Normalizing Flows, Jonas Koehler et al, NeurIPS 2021, [https://arxiv.org/abs/2110.0035](https://arxiv.org/abs/2110.00351)
>
> [2] Equivariant Flow-Based Sampling for Lattice Gauge Theory, Kanwar et al, Physics Review Letters 2020, [https://journals.aps.org/prl/abstract/10.1103/PhysRevLett.125.121601](https://journals.aps.org/prl/abstract/10.1103/PhysRevLett.125.121601)
>
> [3] Sampling using SU(N) gauge equivariant flows, Boyda et al, Physical Review D, [https://journals.aps.org/prd/abstract/10.1103/PhysRevD.103.074504](https://journals.aps.org/prd/abstract/10.1103/PhysRevD.103.074504)
>
> [4] Flow-based sampling in the lattice Schwinger model at criticality, Albergo et al, 2022, Physical Review D [https://journals.aps.org/prd/abstract/10.1103/PhysRevD.106.014514](https://journals.aps.org/prd/abstract/10.1103/PhysRevD.106.014514)
>
> **The authors emphasise that an advantage of their method relative to those from Vaitl et al. for the estimation of the forward KL is that their method does not require reweighting. However, their method uses samples from the target, while the method from Vaitl et al. uses samples from the flow - hence the two methods are not directly comparable as they are for different situations. I think this is somewhat misleadingly presented in the text (it is presented as an improvement relative to the forward KL objective from Vaitl).**
>
> We agree that this is an important difference between our method and the one proposed by Vaitl et al. We have revised the manuscript to make this clearer.
> We stress however that their reweighting method comes with an important downside: it fails as the system size grows because the probability mass of the target density becomes increasingly concentrated, see for example Figure 4 right in Vaitl et al. As a result, the importance weights suffers from large variance.
> Futhermore, their reweighting-based approach does not allow to incorporate samples from MD or MCMC into the training. This is unfortunate as Boltzmann generators are often trained using such samples. We discussed extensively with the authors of Vaitl et al. and they agree with this assesment.
> Even if the bulk of the training is energy-based, samples are essential as a pretraining setp because the self-sampling nature of the energy-based training will fail to find any modes when the flow is randomly initialized and the target distribution is high-dimensional.
> Only our approach allows for path-gradients of sample-based maximum likelihood training. It is therefore a substantial improvement upon the method of Vaitl et al. and of great interest to the Boltzmann generator community.

---

> > ### Author Response · Authors · 2023-11-21
> >
> > ### Questions
> >
> > **How come the flow trained via the standard maximum likelihood objective achieves such poor performance on the MGM problem (Table 1)?. It seems possible that poor hyper-parameters have been used as training by maximum likelihood should be able to obtain reasonable results**
> >
> > Note that our experiments merely establish that path gradients facilitate *more sample efficient training* and help avoid overfitting.
> > This can be seen from the rhs of Fig 1, namely that standard maximum likelihood training can easily fit the MGM if enough samples are provided. For smaller number of samples, maximum likelihood training leads to overfitting. A non-vanishing ESS_p could be achieved by early stopping. Early stopping is however challenging in the context of Boltzmann generators, as their training often relies on a limited number of biased samples which are only used for pre-training. In the revised manuscript, we nevertheless report results with early stopping to adopt the most charitable setting for the baseline. We stress that sample efficiency is a crucial requirement in learning unnormalized distributions for which training set generation involves costly MD or MCMC simulations.
> >
> > To avoid any impression of hyperparameter tuning and to provide a more detailed analysis, we include an extensive analysis of results for various batch and flow sizes in the revised Appendix E. In more detail, Figures 5, 6 and 7 show the corresponding Forward ESS during the course of both path and non-path forward training for varying numbers of
> > - linear layers per coupling block,
> > - number of hidden neurons per linear layer,
> > - batch sizes.
> >
> > While standard Forward KL gradients can *relatively* surpass Forward KL Path Gradients for smallest models trained with the two smallest batch sizes by a small margin, a larger model trained with path gradients doubles the mean of the $ESS_p$ in a direct comparison and improves upon the best forward KL gradients by almost 10 percentage points in absolute terms.
> >
> > Tables 3, 4 and 5 summarize the best possible Forward ESS over the course of training (corresponding to the results obtained by early stopping) for direct comparison.
> >
> > As the additional experiments demonstrate, the superior performance of our method does not depend on the particular hyperparameter choices.
> >
> > **In the case of forwards KL with flows that require implicit differentiation for inversion, is it not more efficient to set the forwards direction of the flow to map from the target to the flow’s base (rather than base to target), such that implicit differentiation is required for sampling, but not density evaluation)?**
> >
> > You are correct in that one can choose the “directionality” of the flow such that density estimation is fast.
> > In such as situation, implicit differentiation is not necessary.
> > However, such a choice is strongly disfavored in the context of Boltzmann generators: one wants to use these flows to facilitate fast sampling from a given target distribution.
> > As such, the sampling direction has to be sufficiently fast. Furthermore, one often combines sample-based pretraining with energy-based training which uses the base-to-target direction of the flow. As a result, sample-based forward training has to rely on implicit differentation. Therefore, our implicit path gradients are of great significance to improve training of Boltzmann generators.

---

> > > ### Comment · Reviewer_UvVP · 2023-11-21
> > >
> > > Thank you for providing the further experiments.
> > >
> > > I think reviewer ZWTi correctly pointed out that the paper's original hyper-parameters on the MGM problem were chosen to accentuate the authors method above the baseline. I think this is quite a big red flag.
> > >
> > > With a smaller batch size and network size indeed the proposed method is no longer better than the alternative.
> > > As the MGM problem is relatively simple, a batch size of 500 and network width of 100 does not seem to be particularly small. It seems like the baseline forward KL method may do better with an even lower batch size.
> > >
> > > I believe that in Figure 1, if a better batch size/early stopping were used for the forward KL baseline that the difference between the curves would be much less dramatic?
> > >
> > > Due to concerns with the accuracy of the presentation and fairness of the experiments I am downgrading my score. Overall, I feel that this paper has significant contributions and would be worth acceptance if the experiments/presentation were more balanced.

---

> > > > ### Author Response · Authors · 2023-11-21
> > > >
> > > > Thank you for your swift reply.
> > > >
> > > > We understand your concern, but kindly refer you to Figures 5 to 7 in the revised manuscript, which address this topic. Those figures summarize the performance of path gradients vs. standard forward training for various batch and flow sizes. These figures clearly establish that your concern does not hold:
> > > > - The best overall performance is obtained by path gradients.
> > > > For the best performing flow sizes, the standard estimator leads to overfitting while our path gradient estimator acts as a regularizer thereby preventing overfitting.
> > > > - The suggested batch size of 500 and network width of 100 only leads to an ESS of 68.9 (well below the best performing model with 91.8 percent which used path gradients).
> > > > - We have provided Figure 5 with Table 3, Figure 6 with Table 4 and Figure 7 with Table 5 for the performance during training and the best possible performance which show improved performance for larger batch sizes and wider linear layers.
> > > > - We are happy to provide results for other values of the batch size and hidden sizes if the reviewer suspects that they could lead to overall better results. In particular, it is higly unlikely that the baseline forward KL would do better than the best ESS 91.8 percent. We are however happy to check if you find this helpful.
> > > >
> > > > We emphasize that overfitting is a serious concern in training Boltzmann generators. This is because samples from the target distribution are costly and thus often a low number of samples need to used for pretraining before switching to energy-based training.
> > > >
> > > > Finally, we understand the general concern of the reviewer and added a sentence to the main paper, referring to the additional experiments in the appendix and making it more clear that performance of course differs for different hyperparameters. Important, however, is that path gradient estimators are often (in >95\% of the configurations) better and never significantly worse than standard estimators.

---

> > > > > ### Comment · Reviewer_UvVP · 2023-11-22
> > > > >
> > > > > Thank you for providing these further experiments - indeed these do ease my concerns over the validity of the results and I have increased my score. Additionally, I am interested as to whether the authors will be making their code publicly available upon publication?

---

> > > > > > ### Author Response · Authors · 2023-11-22
> > > > > >
> > > > > > Thank you for your swift reply and for raising your score. Yes, we will release code after some further polishing.

---

### Official Review · Reviewer_vn2u · 2023-10-31

**Soundness:** 3 good
**Presentation:** 2 fair
**Contribution:** 2 fair
**Rating:** 8
**Confidence:** 3

**Summary:**

This paper considers the problem of learning a distribution $p$ given an oracle
for log probabilities plus a constant (i.e., $\log p(x) + c$ at sample $x$). It
proposes a method for estimating the gradients of forward and reverse KL
divergence that dispenses with a term known to have zero expectation value, thus
allowing lower variance estimators of the gradient, with less computational
complexity than prior work. In particular, this method deployed beyond previous
results for continuous flows to include coupling flows.

**Strengths:**

The paper technically precise and, to my knowledge, presents valuable original
work with immediate applications. The experiments were generally informative.
Its major contribution is reducing the computational complexity for calculating
path gradients of both forward and reverse KL when $\log p(x) + c$ is queriable.

The theoretical results appear sound after some inspection.

I believe the overall contribution is valuable enough to share with the broader
ICLR community, though I was surprised that the proposed "fast" gradient
estimator was not already established. Perhaps like many key results, it seems
obvious in hindsight. The suggestion that removal of the $\frac{\partial}{\partial \theta} \log q$
term from the gradient estimate makes learning empirically robust to overfitting
is quite interesting and provocative, but unexplored in detail.

**Weaknesses:**

I had some difficulty reading this work, despite some prior exposure to the
subject matter. It took me several passes to make sense of what the key
contribution was, and I wished for additional clarity.  The key idea behind
"path gradients" (dropping a term that has zero expectation value) from the
empirical estimation of the gradient is easy enough to understand, but took some
time to distill from the intro [1].

Regarding the experiments, at least one sentence introducing effective sample
size would also have been appreciated.

[1] It took me far too long to realize that the expectation value in Equation
(10) was for $x_0 \sim q_0$, not $x \sim q_{\theta}$. This might have been
more clear if different symbols were used for inputs $x_0 \to x$ and outputs
$x \to y$ of the transformation, since layer indexing was only used in the
context of coupling flows.

**Questions:**

No questions.

---

> ### Author Response · Authors · 2023-11-21
>
> ### Strengths
>
> **The paper technically precise and, to my knowledge, presents valuable original work with immediate applications. The experiments were generally informative. Its major contribution is reducing the computational complexity for calculating path gradients of both forward and reverse KL when log⁡p(x)+c is queriable.**
>
> **The theoretical results appear sound after some inspection.**
>
> **I believe the overall contribution is valuable enough to share with the broader ICLR community, though I was surprised that the proposed "fast" gradient estimator was not already established. Perhaps like many key results, it seems obvious in hindsight. The suggestion that removal of the $\frac{\partial}{\partial \theta}  log(⁡ q )$ term from the gradient estimate makes learning empirically robust to overfitting is quite interesting and provocative, but unexplored in detail.**
>
> We thank the reviewer for the positive feedback. We agree that path gradients lead to more robust training for normalizing flows in the context of tractable ground-truth energy function $\log ⁡p(x)+c$.
>
> ### Weaknesses
>
> **I had some difficulty reading this work, despite some prior exposure to the subject matter. It took me several passes to make sense of what the key contribution was, and I wished for additional clarity. The key idea behind "path gradients" (dropping a term that has zero expectation value) from the empirical estimation of the gradient is easy enough to understand, but took some time to distill from the intro [1].**
>
> We have rephrased the introduction to make this point and our key contributions clearer.
>
> **Regarding the experiments, at least one sentence introducing effective sample size would also have been appreciated.**
>
> We have expanded the discussion of the Effective Sampling Size in appendix E.
>
> **It took me far too long to realize that the expectation value in Equation (10) was for $x_0 \sim q_0$, not $x \sim q_\theta$ This might have been more clear if different symbols were used for inputs**
>
> **$x_0 \to x$ and outputs $x \to y$ of the transformation, since layer indexing was only used in the context of coupling flows.**
>
> We will update the draft following your suggestions for the camera-ready version. We however prefer to use $z=x_0$ as it is more standard notation.

---

### Official Review · Reviewer_yXNA · 2023-11-04

**Soundness:** 3 good
**Presentation:** 3 good
**Contribution:** 3 good
**Rating:** 8
**Confidence:** 3

**Summary:**

This work deals with improving the pathwise gradient
estimator in the context of variational inference
using normalizing flow based models (i.e., they
want a fast method for computing the "sticking the landing"
estimator by Roeder).
In particular, they are looking at deriving pathwise gradients
for the log probability term of the normalizing flow.
Computing this efficiently is non-trivial due to the
modification to the probability caused by a change of coordinates
that requires computing the determinant of the Jacobian.

They derive a faster method for computing the pathwise gradient in
this setting for coupling flows (the most widely used normalizing
flow). The improvement in computational speed ranges between 1.3 times
to 8 times (takes 1.4 - 2.3 times the standard estimator that has a
higher variance, so doesn't work as well). The improvement is
especially large for implicitly invertible coupling flows, but more
modest for explicitly invertible coupling flows.

Their formulation allows computing the pathwise gradient for
both the forward and reverse KL, allowing to also perform
maximum likelihood training.

Experiments were performed on a multimodal Gaussian distribution as
well as physics settings: U(1) gauge theory and $\phi^4$ lattice
model.

**Strengths:**

+Fast pathwise gradients are certainly necessary for normalizing flows,
and the current work provides this with a large improvement over the
prior work in terms of computational speed.

+The method improves in both walltime and efficiency.

+The method allows both forward and reverse KL training.

**Weaknesses:**

-The literature review is a bit misleading, as pathwise
gradients have been around for a long time, e.g., see [L'Ecuyer,
P. (1991). An overview of derivative estimation] where it is
referred to as "infinitesimal perturbation analysis". Moreover,
reparameterization gradients are a type of pathwise gradient, and
there are other works discussing it, e.g., [Jankowiak & Obermeyer, 2018]
or [Parmas & Sugiyama, 2021]. The current work is mainly referring
to pathwise gradients in the context of normalizing flows and
variational modeling, but the broader picture of pathwise gradients
should be briefly mentioned, and probably the terminology should
be clarified because the current paper refers to "pathwise" gradients
as the narrow application of it to normalizing flows, whereas there
are many other estimators that have been around for decades that are
also referred to as pathwise estimators.

-The experiments are a bit toy, or at least their significance
was not explained.

Jankowiak, M., & Obermeyer, F. (2018, July). Pathwise derivatives
beyond the reparameterization trick. In International conference on
machine learning (pp. 2235-2244). PMLR.

Parmas, P., & Sugiyama, M. (2021, March). A unified view of likelihood
ratio and reparameterization gradients. In International Conference on
Artificial Intelligence and Statistics (pp. 4078-4086). PMLR.

**Questions:**

I have a naive question about computing the pathwise gradient of the
reverse KL. In equation (2), it seems to me that we could rewrite the
equation by using the Jacobian of the forward transform based on the
inverse function theorem, so that the $+\log |\textup{det} ~ dT^{-1}/dx|$ term
becomes $- \log |\textup{det}~dT/dx_0|$. Then we could compute the quantity and
use backprop to get the pathwise gradient. Am I misunderstanding, or
why would this not work? Is the computation of the Jacobian too
costly?

"Path gradients have the appealing property that they are unbiased and
have lower variance compared to standard estimators, thereby promising
accelerated convergence (Roeder et al., 2017; Agrawal et al., 2020;
Vaitl et al., 2022a;b)."  -> Other estimators are also unbiased. The
sentence makes it seem like they aren't. Also, the "have lower
variance" is not always true. I suggest revising to make the sentence
correct, e.g., making it "tend to have lower variance".

---

> ### Author Response · Authors · 2023-11-21
>
> ### Strengths
>
>
> - **Fast pathwise gradients are certainly necessary for normalizing flows, and the current work provides this with a large improvement over the prior work in terms of computational speed.**
> - **The method improves in both walltime and efficiency.**
> - **The method allows both forward and reverse KL training.**
>
> We thank the reviewer for pointing out these strenghts. We agree that that path gradients are an important tool in successfully training normalizing flows.
>
> ### Weaknesses
>
> **The literature review is a bit misleading, as pathwise gradients have been around for a long time, e.g., see [L'Ecuyer, P. (1991)]. An overview of derivative estimation] where it is referred to as "infinitesimal perturbation analysis". Moreover, reparameterization gradients are a type of pathwise gradient, and there are other works discussing it, e.g., [Jankowiak & Obermeyer, 2018] or [Parmas & Sugiyama, 2021]. The current work is mainly referring to pathwise gradients in the context of normalizing flows and variational modeling, but the broader picture of pathwise gradients should be briefly mentioned, and probably the terminology should be clarified because the current paper refers to "pathwise" gradients as the narrow application of it to normalizing flows, whereas there are many other estimators that have been around for decades that are also referred to as pathwise estimators.**
>
> We thank the reviewer for pointing this out. We indeed focused narrowly on normalizing flow applications in the literature review. In light of the reviewers comments, we have extended the discussion with the suggested references providing a broader perspective on the field.
>
> **The experiments are a bit toy, or at least their significance was not explained.**
>
> We politely disagree with this statement. Lattice field theory provides the mathematical framework underlying many parts of modern theoretical physics, in particular, high-energy physics, gravitational physics, condensed matter and statistical physics. Indeed all known fundamental forces of nature can be described by quantum field theory (gravity only in an effective field theory sense). Applications of normalizing flows to lattice field theory is an emerging and highly promising field, see, for example, this recent review ([https://www.nature.com/articles/s42254-023-00616-w](https://www.nature.com/articles/s42254-023-00616-w)). In fact, a significant part of the global supercomputing resources are spent on lattice field theory simulations.
>
> From a machine-learning point of view, this problem setting is extremely challenging because of the extremely large symmetry of the target distribution. Specifically, the energy (or, more precisely, action) of many lattice field theories is invariant under local symmetry groups, i.e., the field at each lattice site can be rotated with a separate symmetry operation. As such the size of the symmetry group scales with the number of lattice sites. Without making the model architecture manifestly invariant under this symmetry, the generative model would not be able to learn. Furthermore, lattice field theories have to be extrapolated to the continuum limit. In other words, they have to be considered at a second order phase transition for which it is well known that the correlation length of the system diverges. Therefore, the generative model has to learn long-ranged interactions.
>
> We also emphasize that our experiments considered state-of-the-art architectures for both $\phi^4$ and U(1) gauge theory. Thus, our manuscript demonstrates improvements of models that are at the forefront of this exciting and highly dynamic field of research.

---

> > ### Author Response · Authors · 2023-11-21
> >
> > ### Questions
> >
> > **I have a naive question about computing the pathwise gradient of the reverse KL. In equation (2), it seems to me that we could rewrite the equation by using the Jacobian of the forward transform based on the inverse function theorem, so that the $+ \log |\det dT_{\theta}^{-1}/dx|$ term becomes $- \log |\det dT_{\theta}/d x_{0}|$. Then we could compute the quantity and use backprop to get the pathwise gradient. Am I misunderstanding, or why would this not work? Is the computation of the Jacobian too costly?**
> >
> > The stated identity is, of course, correct but unfortunately unhelpful as we are interested in the derivative with respect to $x,$ i.e.,   $\frac{\partial \log \det |\frac{d T_{\theta}^{-1}(x) }{ d x }|}{\partial x}$. The term $\log |\det dT_{\theta}/d x_{0}|$ manifestly only depends $x_0$ and only implictly on $x$ through $x_0 = T^{-1}_{\theta}(x)$. Of course, the gradient can straightforwardly be calculated using the chain rule
> >
> > $\frac{\partial \log \det |\frac{d T_{\theta}^{-1}(x)}{d x}|}{\partial x} = - \frac{\partial \log \det |\frac{d T_{\theta}(x_0)}{d x_0}|}{\partial x_0} \frac{\partial T_{\theta}^{-1}(x)}{\partial x}.$
> >
> > However this involves a Jacobian in the target-to-base direction which we wanted to avoid in the first place.
> >
> > **"Path gradients have the appealing property that they are unbiased and have lower variance compared to standard estimators, thereby promising accelerated convergence (Roeder et al., 2017; Agrawal et al., 2020; Vaitl et al., 2022a;b)." -> Other estimators are also unbiased. The sentence makes it seem like they aren't. Also, the "have lower variance" is not always true. I suggest revising to make the sentence correct, e.g., making it "tend to have lower variance".**
> >
> > We fully agree and have rephrased the relevant sentences.

---

> > > ### Comment · Reviewer_yXNA · 2023-11-23
> > >
> > > Thanks a lot for the comment and the clarification regarding physical significance.
> > > I also looked at the discussion with other reviewers, and I remain positive about the paper.
> > > I think it's a strong paper and should be accepted.
> > >
> > > Just one more clarifying question regarding the gradient computation to test my understanding.
> > > It seems to me that you do not need the Jacobian in the target-to-base direction (as you claimed), but only the Jacobian-vector product.
> > > One could create the computation graph from the target-to-base, then set the output gradients to the gradient w.r.t. $x_0$, and compute the required quantity using backprop. Is there some reason why this is not promising/does it correspond to any of the baselines? Actually, looking at the paper again, this seems to be the method of Vaitl that you have detailed in Section 3. Please correct me if I misunderstood something.
> > >
> > > Anyhow, I think this is a strong paper. Thank you for the clarifications.

---

> > > > ### Comment · Area_Chair_7fU3 · 2023-12-05
> > > > **Forwarding the response message from authors**
> > > >
> > > > Dear Reviewer yXNA,
> > > >
> > > > The authors could not respond to your last comment before the end of discussion period and sent the response message to me. I think it is appropriate to forward this message to you for your reference.
> > > >
> > > > "Thank you for your reply and your positive assessment of our manuscript. We unfortunately did not manage to reply before the discussion period ended. We hope this reply is useful. You are correct about the vector Jacobian product. The method that you describe indeed closely corresponds to Vaitl et al. Note however that they calculate \frac{\partial log det J}{\partial x} \frac{\partial T_\theta(x_0}{\partial \theta}, i.e., the full path gradient, and not the derivative wrt to x. But the idea is essentially the same."
> > > >
> > > > Thanks,
> > > >
> > > > Area Chair 7fU3

---

### Author Response · Authors · 2023-11-21

Dear reviewers,

thank you very much for your thorough and insightful review. We sincerely appreciate the time you spend in evaluating our work and your comments which helped us to improve our manuscript considerably. We are glad you appreciate the ideas of the paper and its relevance to the normalizing flow community.

### Summary

Recent work has established the superiority of path gradient estimators for the training of normalizing flows on the Reverse KL.
In this work we propose a fast path gradient estimator which works for all normalizing flow architectures of practical relevance for sampling from an unnormalized target distribution. We show that this estimator can also be applied to previously unaddressed maximum likelihood training and empirically demonstrate its superior performance for several natural sciences applications.

### Contribution

The reviewers agreed on the usefulness, improved speed and applicability of fast path gradients for training normalizing flows in the context of Boltzmann generators which is of high relevance for ML4Science.
Additionally, all reviewers were of the opinion that the training with path gradients in both direction, whether base to target via the Reverse KL or target to base distribution via the Forward KL is a key contribution.

### Adressed Concerns

- We extended the related works section incorporating relevant references pointed out by the reviewers. Specifically, we give a broader perspective on path gradients beyond their applications to normalizing flows.

- One reviewer suggested to include the recursive path gradient formula for general normalizing flow architectures and then to specialize to both implicitly and explicitly invertible coupling flows. This allows to more clearly highlight the specific cost advantages of our proposal. We implemented this suggestion adding a new proposition in the main text for the generic case. Furthermore, we added discussions detailing the runtime advantages of our approach for coupling flows. We also clarified why the Jacobian of the coupling transformation is square and invertible in both Proposition 3.2 and its proof.

- We significantly extended Appendix B.3 to give further details on the regularizing properties and reduced variance of our forward KL path gradient. Specifically, we also added Figure 3 (analogous to the one for the reverse KL by Vaitl et al.) illustrating its sticking-the-landing property.

- We have updated the introduction to emphasize the key conceptual idea underlying path gradient: a term that vanishes in expectation is dropped in the gradient estimation.
- Concerning the experiments, two reviewers wondered if the low forward effective sampling size ESS_p for the baseline was due to a particular choice of hyperparameters, such as batch and model size. We clarify this in the revised manuscript by i) reporting results with early stopping in the tables and ii) substantially extending Appendix E with an results summarizing an extensive sweep over hyperparameters. Again, path gradients outperformed the baseline in 95% of the tested configurations.

For convenience, we have highlighted all changes made to the manuscript in blue.

---

### Meta-Review · Area_Chair_7fU3 · 2023-12-05

**Metareview:**

The paper developed a more efficient path gradient estimation method for energy-based training of normalizing flows. The method is developed by zooming into the path gradient calculation process to reuse the evaluation pass and walk around inverse flow evaluation, which is especially helpful for implicit invertible flows (which trade easy inverse evaluation for better expressiveness). The authors also applied the method to optimizing the forward KL loss that leverages data, which makes an inverse-evaluation-free estimator with lower variance. The method is well motivated and solid, and experiments showed significant speedup for implicit inverse flows and improved learning performance on a few natural science problems.

**Justification For Why Not Higher Score:**

Although the investigated distribution learning problems in natural science bare their own values and difficulties, it would attract interest from a broader audience if applicability to more general AI tasks is shown. The forward KL version also requires the energy function in addition to data samples, which restricts the applicability to general maximum likelihood training scenarios. The method may require more cost for general flow-based models other than coupling flows.

**Justification For Why Not Lower Score:**

All reviewers appreciated the solidity and value of the proposed method for enhancing the applicability of flow-based models. Reviewers UvVP and ZWTi found concerns regarding experimental settings and results, which are subsequently resolved by authors' additional more detailed experiments during the rebuttal period.

---

### Decision · Program_Chairs · 2024-01-16

Accept (poster)